# Correlation of GAA Genotype and Acid-α-Glucosidase Enzyme Activity in Hungarian Patients with Pompe Disease

**DOI:** 10.3390/life11060507

**Published:** 2021-05-31

**Authors:** Aniko Gal, Zoltán Grosz, Beata Borsos, Ildikó Szatmari, Agnes Sebők, Laszló Jávor, Veronika Harmath, Katalin Szakszon, Livia Dezsi, Eniko Balku, Zita Jobbagy, Agnes Herczegfalvi, Zsuzsanna Almássy, Levente Kerényi, Maria Judit Molnar

**Affiliations:** 1Institute of Genomic Medicine and Rare Disorders, Semmelweis University, 1082 Budapest, Hungary; gal.aniko@med.semmelweis-univ.hu (A.G.); grosz.zoltan@med.semmelweis-univ.hu (Z.G.); borsos.beata@med.semmelweis-univ.hu (B.B.); 2First Department of Pediatrics, Semmelweis University, 1082 Budapest, Hungary; szatmari.ildiko@med.semmelweis-univ.hu; 3Department of Neurology, University of Pecs, 7622 Pecs, Hungary; sebok.agnes@pte.hu; 4Department of Neurology, Aladar Petz University Teaching Hospital, 9023 Győr, Hungary; intezmenyvezeto@petz.gyor.hu; 5Department of Pediatrics, St. Rafael Hospital of Zala County, 8900 Zalaegerszeg, Hungary; harmathvera@gmail.com; 6Department of Pediatrics, Faculty of Medicine, University of Debrecen, 4032 Debrecen, Hungary; szakszon.katalin@gmail.com; 7Department of Neurology, University of Szeged, 6720 Szeged, Hungary; dezsi.livia@med.u-szeged.hu; 8Department of Pediatrics, Andras Josa Teaching Hospital, 4400 Nyiregyhaza, Hungary; dr.balku.eniko@josa.hu; 9Department of Neurology, Bács-Kiskun County Hospital, 6000 Kecskemét, Hungary; zitajobbagy@hotmail.com; 10II. Department of Pediatrics, Semmelweis University, 1082 Budapest, Hungary; herczegfalvi.agnes@med.semmelweis-univ.hu; 11Department of Toxicology, Heim Pal Children’s Hospital Budapest, 1089 Budapest, Hungary; almassyzs@gmail.com; 12Department of Neurology, Szent György County Hospital, 8000 Székesfehérvár, Hungary; lkerenyi@mail.fmkorhaz.hu

**Keywords:** Pompe disease, GAA genotype, GAA enzyme activity

## Abstract

Pompe disease is caused by the accumulation of glycogen in the lysosomes due to a deficiency of the lysosomal acid-α-glucosidase (GAA) enzyme. Depending on residual enzyme activity, the disease manifests two distinct phenotypes. In this study, we assess an enzymatic and genetic analysis of Hungarian patients with Pompe disease. Twenty-four patients diagnosed with Pompe disease were included. Enzyme activity of acid-α-glucosidase was measured by mass spectrometry. Sanger sequencing and an MLPA of the GAA gene were performed in all patients. Twenty (83.33%) patients were classified as having late-onset Pompe disease and four (16.66%) had infantile-onset Pompe disease. Fifteen different pathogenic GAA variants were detected. The most common finding was the c.-32-13 T > G splice site alteration. Comparing the α-glucosidase enzyme activity of homozygous cases to the compound heterozygous cases of the c.-32-13 T > G disease-causing variant, the mean GAA activity in homozygous cases was significantly higher. The lowest enzyme activity was found in cases where the c.-32-13 T > G variant was not present. The localization of the identified sequence variations in regions encoding the crucial protein domains of GAA correlates with severe effects on enzyme activity. A better understanding of the impact of pathogenic gene variations may help earlier initiation of enzyme replacement therapy (ERT) if subtle symptoms occur. Further information on the effect of GAA gene variation on the efficacy of treatment and the extent of immune response to ERT would be of importance for optimal disease management and designing effective treatment plans.

## 1. Introduction

Glycogen storage disease type II (Pompe disease or acid maltase deficiency) is an autosomal recessive metabolic disorder characterized by the accumulation of glycogen in the lysosomes due to a deficiency of the lysosomal acid alpha-glucosidase (GAA) enzyme. The gene encoding the enzyme is located on chromosome 17q25.2-q25.3. Pompe disease has two distinct types, the infantile (IOPD) and late-onset (LOPD) forms [1,2,3]. The classical infantile form is characterized by severe hypotonia, cardiomyopathy and respiratory insufficiency in the first months after birth, leading to death in one year if left untreated. In non-classical IOPD, the cardiac involvement is less severe [4]. The late-onset form can manifest from childhood through late adulthood and limb-girdle muscle weakness accompanied by respiratory insufficiency is common [5,6,7]. The clinical phenotypes are associated with the levels of residual GAA enzyme activity. Less than 3% of normal enzyme activity is usually detected in the severe classical infantile-onset cases, while residual levels ranging between 3% and 30% of normal value are found in the less severe late-onset form [6,7]. Lower α-glucosidase activity can also be the result of the presence of an allele, resulting in pseudodeficiency. A pseudodeficient allele never causes Pompe disease, this is one of the reasons why genetic testing must be performed during the diagnostic process of Pompe disease [8]. Patients are classified into CRIM (cross reactive immunologic material) negative and CRIM positive groups, depending on whether the enzyme is completely absent (CRIM-) or present in lower amounts (CRIM+) [9]. Pompe disease has been treated by the regular substitution of recombinant GAA enzyme since 2006, which significantly improved survival and reduced the severity of symptoms in patients of both subtypes. However, it was found to be more effective on reversing cardiac symptoms (one of the main contributors of mortality in the classical IOPD) than on skeletal muscle dysfunction. Enzyme replacement therapy (ERT) is safe and well tolerated [10,11,12,13].

Until now, according to the ClinVar database (http://www.ncbi.nlm.nih.gov/clinvar), there were over 372 pathogenic or likely pathogenic alterations, and 541 variations with uncertain significance are known in the GAA gene, which result the absence or diminished activity of the enzyme. Decreased enzyme activity leads to impaired degradation of glycogen to glucose in the lysosomes. In the early stage, cells contain lysosomes filled with glycogen. As the disease progresses, lysosomes expand in size and number and the membrane of lysosomes rupture, allowing the glycogen to enter the cytoplasm of the cell. The cell flooded with glycogen can no longer maintain its basic processes and dies [14]. The underlying pathophysiological cascade involves the disrupted mechanism of autophagy, protein misfolding and impaired intracellular trafficking, leading to the generation of lipofuscin, an insoluble brownish colored degradation product of unsaturated fatty acids and glycoproteins cross-linked with metal ions [15,16,17,18,19,20]. Generally, widespread accumulation of lipofuscin is recognized as a result of oxidative stress and accompanies the phenomenon of aging [21]. In Pompe disease, beside the primary lysosomal pathology, large amounts of non-contractile lipofuscin inclusion and autophagosomal buildup were found in the muscle fibers of LOPD patients, contributing to their clinical status. These pathological findings showed progression over time, even in patients on ERT treatment [20]. A muscle biopsy of a patient after 6 years on ERT treatment revealed the clearance of intralysosomal glycogen, but this did not affect the progressive accumulation of these debris. The greater the extent and number of autophagosomal and lipofuscin inclusions are before ERT treatment, the less clinical improvement is achieved [20]. This finding indicates that earlier disease onset and later-initiated ERT favors the accumulation of secondary pathological materials, which impair the clinical response to treatment [20].

In this study, we publish the results of the first enzymology and genetic analysis of a Hungarian cohort with Pompe disease. We assessed the protein domain involvement linked to different disease-causing alterations with the possible effect these variants may have on protein structure, GAA enzyme activity and phenotypes.

## 2. Materials and Methods

### 2.1. Studied Cohort

Twenty-four Hungarian patients from 19 families with Pompe disease were enrolled (16 females and 8 males). The age of participants (in 2021) ranged from 4 to 69 years (mean age 43.5 ± 21.1 years (CI 95% 34.6–51.4)). Patients had Hungarian ethnicity. Written informed consent was obtained from the patients and controls before the sample collection and molecular genetic testing. The study was approved by the Ethical Committee of Semmelweis University. Molecular genetic analysis was performed for diagnostic purposes in all investigated patients. Some of the genetic investigations were performed in the Laboratory of ARCHIMED Life Science (ARCHIMED Life Science, Vienna, Austria).

### 2.2. Molecular Genetic Analysis of GAA Gene

DNA was extracted from blood using the QIAamp DNA blood kit, according to the manufacturer’s instructions (QIAgen, Hilden, Germany). The total coding region of the GAA gene was analyzed by Sanger sequencing using ABI Prism 3500 DNA Sequencer (Applied Biosystems, Foster City, CA, USA). The sequence was compared to the human reference genome (GRCh38.p13, ENST00000302262.7, NM_000152) using NCBI’s Blast^®^ application. Quantitative changes in the GAA gene were analyzed by a multiplex ligation-dependent probe amplification assay (MLPA) (SALSA MLPA P453 GAA probemix, MRC Holland, Amsterdam, The Netherlands). For the normalization, 3 different healthy control samples were used per each run.

### 2.3. In Silico Analysis

In silico analyses were performed with the Varsome database [22]. The significance of detected alterations was tested with HGMD (www.hgmd.cf.ac.uk), the Exome Aggregation Consortium (ExAC) database (http://exac.broadinstitute.org), 1000Genome (http://www.1000genomes.org), NHLBI Exome Sequencing Project (ESP) (http://evs.gs.washington.edu/EVS/), ClinVar (http://www.ncbi.nlm.nih.gov/clinvar), dbGAP (http://www.ncbi.nlm.nih.gov/gap), EGA (http://www.ebi.ac.uk/ega), dbSNP (www.ncbi.nlm.nih.gov/SNP/) and the Pompe disease GAA variant database (http://www.pompevariantdatabase.nl) [23]. The nature of alterations was established according to the ACMG guideline [24].

### 2.4. GAA Enzyme Activity Measurement

Enzymatic activity of acid-α-glucosidase (GAA) was measured from dried blood spot (DBS) samples using a multiplex assay (five lysosomal enzyme activities—β-glucocerebrosidase (ABG), α-galactosidase A (GLA), galactocerebrosidase (GALC) and α-iduronidase (IDUA)), as described earlier [25,26]. In all cases 3, biological replicates were evaluated from 3 different blood spots. All blood spots were measured 3 times (technical replicates). Flow injection-MS/MS was performed on a Sciex API4000-QTRAP instrument. The Sciex Analyst Instrument control and Data processing software package (1.4.1) was used to integrate all product and internal standard multiple reaction monitoring (MRM) peaks. Performance of the assay was evaluated using CDC quality control materials [27]. Preparation, storage and shipping of DBSs was carried out in line with previous precautions [28]. Decreased α-glucosidase enzyme activity was considered in case of values < 2.0 μmol/L/h. The 5th percentile was calculated based on the total number of samples previously measured by this multiplex assay.

### 2.5. Determination of CRIM Status

In the case of P23 patient, CRIM status was determined on a blood sample for routine diagnostic purpose by Western blotting in the Laboratory of Department of Chemical Pathology, Great Ormond Street Hospital for Children NHS Trust. The laboratory is accredited by UKAS by ISO15189:2012 [29].

### 2.6. Statistical Analysis

The group comparisons were performed with Shapiro–Wilk test, Pearson correlation and one-way ANOVA regarding means. *p* values of <0.05 were considered statistically significant. The 95% confidence intervals (CI95%) for proportions and means were calculated using standard formulas.

## 3. Results

### 3.1. Clinical Assessment

From the 24 studied Pompe patients, 20 (83.33%) were classified as LOPD and four (16.66%) as IOPD (P15 as non-classical IOPD). The patients belonged to 19 different families. In the LOPD cohort, the age of onset ranged from 3 to 64 years (mean age 48.3 ± 17.7 years (CI 95% 40.7–55.3)). The IOPD phenotype was present in three female cases and one male case. The mean age of onset in this subgroup was 4.5 ± 1.8 (CI 95% 2.8–5.1) months, within the range of 3–6 months (0.25–0.5 years).

### 3.2. Genetic Testing and Distribution of GAA Genotype

Fifteen different pathogenic or likely pathogenic GAA variations were detected in the 24 patients in homozygous or compound heterozygous form. A total of 14 GAA genotypes were confirmed (Table 1 and Table 2). The distribution of the different pathogenic variants was the following: 25 (52%) splice site, 15 (31.25%) missense, 5 (10.5%) INDEL, 3 (6.25%) nonsense (Table 2). The c.-32-13 T > G (rs386834236) splice site alteration was present in 20 patients, on 25 affected alleles. Exon duplication or deletion of the GAA gene was not confirmed in any of our cases. Five patients were homozygous, while other 15 were compound heterozygous for the c.-32-13 T > G alteration, accompanied with other disease-causing variants in the later cases (Table 1). All of these cases were associated with the LOPD phenotype. The mean age of onset in homozygous cases was higher than in the compound heterozygous ones (40.4 ± 17.7 vs. 26.9 ± 14.5) (Table 1), but this difference was not statistically significant (*p* = 0.102). The c.925 G > A (p.Gly309Arg) (rs543300039) missense variant was found in compound heterozygous form with the c.-32-13 T > G (*n* = 3) and c.1468 T > C (p.Phe490Leu) variants in four LOPD cases linked to three families. The age of onset was relatively variable, ranging between 3 and 30 years of age. The c.525delT (p.Glu176ArgfsTer45) (rs386834235) frameshift variant was detected in compound heterozygous form with c.-32-13 T > G splice site alteration in three female cases from two families. The age of onset showed close correlation (in the range of 35–40 years) (Table 1). The c.1799 G > A (p.Arg600His) (rs377544304) missense variant was found in three patients in three different families. The accompanying second alteration in two cases was the c.-32-13 T > G non-coding variant. One of them (P14) can be classified as childhood-onset LOPD since his clinical symptoms began at 5 years of age. The clinical symptoms of the other patient (P15) with the same genotype began at the age of 4.5 months, and notable cardiomyopathy was present, but enzyme activity was in the range of LOPD group’s. He classifies as non-classical IOPD (Table 1). In the third patient, the c.1799 G > A missense variant was combined with the c.875 A > G (p.Tyr292Cys) (rs1057516600) missense alteration. This patient is characterized as an infantile-onset case with a very early (3 months old) disease manifestation (Table 1 and Table 2). Another eight pathogenic alterations (c.307 T > G (p.Cys103Gly); c.1465G > A (p.Asp489Asn); c.1562 A > T (p.Glu521Val); c.1564 C > G (p.Pro522Ala); c.1927 G > T (p.Gly643Trp); c.1942 G > A (p.Gly648Ser); c.2269 C > T (p.Gln757Ter); and c.2407 C > T (p.Gln803Ter)) occurred in one family each (Table 1). Furthermore, one novel homozygous inframe variant (c.1158_1160delGGT (p.Val388del) has been identified. This alteration is located at highly conserved protein residues. According to ACMG guidelines, the clinical significance of this variant is considered as likely pathogenic (Table 2) and CRIM status was determined as positive.

According to the Pompe Disease Database (http://www.pompevariantdatabase.nl) [23], the CRIM status of eight variants is positive (c.-32-13 T > G, c.307 T > G, c.875A > G, c. 925 G > A, c.1465 G > A, c.1468 T > C, c.1564 C > G, c.1799 G > A), while two variants are linked to negative CRIM status (c.525delT, c.2269 C > T) (Table 2). According to the Pompe Disease Database, in ten cases, the phenotype with null allele is characteristic of the classic infantile type (Table 2). The most common c.-32-13 T > G splice site alteration is associated with childhood- or adult-onset forms (Table 2). The prediction of the disease-causing alteration to the severity of the phenotype was ‘very severe’ in three cases, ‘potentially less severe’ in six cases, ‘less severe’ in one case, while ‘potentially mild’ in two cases (Table 2).

### 3.3. Correlation between α-Glucosidase Enzyme Activity and GAA Genotypes

The mean α−glucosidase enzyme activity in the IOPD group (0.346 ± 0.341 μmol/L/h) (CI 95% 0.149–0.543) was significantly lower when compared to the LOPD (0.551 ± 0.368 μmol/L/h) (CI 95% 0.442–0.66) cohort, as expected (*p* < 0.01) (Figure 1A–C) (normal range of α−glucosidase enzyme activity is >2.0 μmol/L/h). We examined the possible effect of the pathogenic alterations and their combinations to each domain of the GAA protein and their correlation with the activity of α−glucosidase enzyme. In addition to the most common splice site alteration (c.-32-13 T > G), the other 14 variants are linked to four different domains of the GAA protein (Figure 2A). Most of the variants (*n* = 8) are found to alter the catalytic CH31 (α/β-)8 barrel domain, three variants the N-terminal β-sheet domain, two the proximal β-sheet domain and one is found to alter the Trefoil type-P domain (Figure 2A). Next, we evaluated the relation between the GAA enzyme activity and domain involvement the alterations affected. By comparing the α−glucosidase enzyme activity of the c.-32-13 T > G homozygous and compound heterozygous cases, the mean GAA activity in the homozygous form is significantly higher than that of the compound heterozygous cases (*p* = 0.003) ([c.-32-13 T > G/c.-32-13 T > G]—0.964 ± 0.357 (μmol/L/h) (CI 95% 0.521–1.407), [c.-32-13 T > G/others] –0.550 ± 0.331 (μmol/L/h) (CI 95% 0.367–0.733) (Figure 1A and Figure 2B). If these compound heterozygous cases are listed according to protein domain specificity, a significant difference can be observed regarding the genotypes. The [c.-32-13 T > G/TTPD] variants (P6, P7, P8, P9) linked to lower GAA activity (mean: 0.367 ± 0.293 (μmol/L/h) (CI 95% 0.181–0.553) (*p* = 0.001)) can be compared to the [c.-32-13 T > G/CCBD] variants (P13, P14, P15, P16, P17) (GAA activity is: 0.454 ± 0.325 (μmol/L/h) (CI 95% 0.247–0.661) (*p* = 0.004)) (Figure 2B; Table 1). The lowest enzyme activity in the LOPD group was found with the [NTBD/CCBD] ([c. 925 G > A/c.1468 T > C]) genotype (GAA activity is: 0.155 ± 0.179 (μmol/L/h) (CI 95% 0.12–0.44) (Figure 1B,C).

Next, the correlation between GAA genotype, GAA activity and age of onset in both LOPD and IOPD groups was evaluated. In the LOPD group, very similar age of onset and GAA activity values were found with the [c.-32-13 T > G/c.525delT] and [c.-32-13 T > G/c.2269 C > T] genotypes (Figure 1B). In case of the [c.-32-13 T > G/c.525delT] genotype in all three affected subjects (P7, P8, P9), closely clustered values were indicated for both age of onsets and GAA activities (age of onsets are: 35 ys, 40 yrs. and 35 yrs.; GAA activities are: 0.5, 0.5 and 0.38 μmol/L/h, respectively). In P18 and P19 patients, the genotype [c.-32-13 T > G/c.2269 C > T] was associated with a very similar age of onset (33 and 35 yrs.), but less close GAA activity values. The trends observed in the two genotypes ([c.-32-13 T > G/c.525delT] and [c.-32-13 T > G/c.2269 C > T]) did not show any significant difference using the Pearson correlation (Figure 1B).

The compound heterozygous [c.-32-13 T > G/c.925 G > A] combination was found in three (P10, P11, P12) patients belonging to two different families (F7, F8) (Table 1). In the F7 family, the disease manifested in childhood (age of onset was 3 yrs. and 10 yrs., respectively) but, in patient P12, the same compound heterozygous alteration has been associated with the LOPD phenotype. To be noted, in women (P11, P12), the symptoms started later (Table 1). The [c.-32-13 T > G/c.1799 G > A] GAA genotype was found in two patients (P14, P15) from two different families (F10, F11) (Table 1). One of the patients (P15) is classified as non-classical IOPD, while the other one (P14) as LOPD. In both patients the α−glucosidase activities were very similar (Figure 1B,C; Table 1) but they differ in phenotype, which may underlie the effects of disease modifier factors. In the IOPD cohort, the earliest age of manifestation (3 months) was found in P21 with [c.875 A > G/c.1799 G < A] genotype (Figure 1C, Table 1). Unfortunately, owing to the limited number of cases in the IOPD group, we do not have the statistical power to arrange any associations between the age of onset, GAA activity and GAA genotypes.

## 4. Discussion

This is the first comparison of GAA genotype related to enzyme activity and phenotype in Hungarian Pompe patients. Pompe disease is a panethnic autosomal recessive lysosomal storage disorder due to mutations in the acid alpha-glucosidase (GAA) gene encoding the lysosomal GAA enzyme. Pathogenic GAA variants lead to the deficient activity of GAA enzymes, resulting impaired glycogen degradation and accumulation of glycogen within the lysosomes [14]. We examined the GAA variants and their corresponding genotypes occurring in a Hungarian Pompe cohort. We also analyzed the effect of these genotypes on disease manifestation and the potential alteration they make on the GAA enzyme structure, resulting decreased enzyme activity. Consistent with the results reported in the literature, in our cohort, the GAA enzyme activity measured in the IOPD group is significantly lower compared to the LOPD cases (Figure 1A–C; Table 1) [30,43,44]. Fifteen different pathogenic or likely pathogenic variants were found in 24 patients. Of these, fourteen are previously reported in the literature, while one of them (P23) is a new inframe variant (c.1158_1160del GGT (p.Val388del) detected in homozygous form. This INDEL variant is currently not included in the population databases nor in our own in-house database. The common c.-32-13 T > G (rs386834236) splice site variant was present in 20 patients, affecting 25 alleles, having an allele frequency of 52% (Table 1 and Table 2). Among the investigated patients, this alteration occurred mostly in the LOPD cohort. The c.-32-13T > G splice site variant is very common in patients of Caucasian origin in both LOPD and IOPD types, with the allele frequency ranging from 40% to 70% in different populations. Almost 90% of patients are affected by this alteration on at least one allele [31,45]. Homozygosity for the c.-32-13T > G variant is associated with the classical adult phenotype spectrum [46]. The c.-32-13T > G splice site alteration is located 13 nucleotides upstream of the canonical acceptor splice site of GAA intron 1 [47,48]. Based on functional studies on patient-derived primary fibroblast cells, the main functional effect of this alteration contributes in the pathogenicity driven by the synthesis of different aberrant splicing variants in which exon 2 is partially or completely spliced out and a limited amount of normally spliced GAA mRNA arises [45]. The c.-32-13T > G alteration interferes by binding the splicing factor U2AF65 to the GAA pre-mRNA, almost completely abrogating its interaction with the polypyrimidine tract of exon 2 leading to the general inefficiency of the splicing process. The overexpression of specific mRNA binding proteins can modulate the expression of normally spliced GAA mRNA from the c.-32-13T > G mutated allele [49]. By affecting the overall splicing efficiency, the balance between the GAA splicing isoforms shifts towards the non-functional, exon 2-skipped variants, yet not completely preventing the expression of the normal transcript that can be translated into an enzymatically active GAA protein. Hence, it culminates to variable levels of GAA residual activity, which may explain the variable onset of symptoms and broad spectra of disease manifestation [35,37,45,46,48,50].

Based on our data, the GAA enzyme activity in c.-32-13T > G homozygous cases are significantly higher when compared to cases with the c.-32-13T > G alteration is present in compound heterozygous form with another pathogenic variant. Similar results were obtained in newborn GAA screening reported in Pennsylvania, where enzyme activity of GAA was significantly higher in homozygous cases compared to compound heterozygous ones [30]. This difference may be due to the fact that, in c.-32-13T > G homozygous cases, only exon 2 of GAA gene is misspliced while, in cases of other variants, the protein may be truncated or its conformation may be significantly altered. However, further functional and protein structural studies would be needed to elucidate the exact pathomechanism behind this phenomenon. Relying on the literature data and the Pompe Disease Database (http://www.pompevariantdatabase.nl) [23], the c.-32-13T > G alteration is linked to both IOPD and LOPD forms [43]. In our cohort, this mutation was found dominantly in LOPD cases (19 LOPD and 1 IOPD) (Table 1; Figure 1A–C and Figure 2A,B).

There is only one patient (P22) in our cohort belonging to the LOPD group who does not have the most common c.-32-13T > G alteration (Table 1). In her case, the disease is characterized by earlier onset (AOO 23 years) and relatively low enzyme activity (0.16 μmol/L/h) (Table 1, Figure 1A,B). Compared to a comprehensive study in Pennsylvania, where 21 LOPD patients featured the c.-32-13T > G alteration, the mean GAA activity of patients did not differ significantly from the group carrying the c.-32-13T > G variant at least on one allele [30]. One of our patient was compound heterozygous for the c.925 G > A (p.Gly309Arg) and the c.1468 T > C (p.Phe490Leu) alteration involving the N-terminal-β-sheet and catalytic CH31 (α/β)8 barrel domains (Table 1, Figure 1A). The c.925G > A (p.Gly309Arg) variant has been reported in at least five patients with Pompe disease [36,51,52]. Expressed in COS-7 cells, this variant did not produce a decreased GAA activity and showed evidence of abnormal processing [34,53]. The c.1468 T > C (p.Phe490Leu) alteration is currently not reported in some of the public databases such as ClinVar or HGMD. Based on the Pompe database, this alteration is associated with ‘less severe’ prediction and positive CRIM status (http://www.pompevariantdatabase.nl) [23]. Among our patients in both the LOPD and IOPD groups, the c.1799 G > A (p.Arg600His) variant was associated with relatively early disease manifestation (Table 1, Figure 1B,C). This alteration was found in one LOPD patient and two IOPD patients. The second alteration in the LOPD case (P14) is c.-32-13 T > G, with clinical symptoms beginning in early childhood (5 years). In the IOPD patients, this variant was combined with the c.-32-13 T > G and the c.875 A > G (p.Tyr292Cys) (rs1057516600) disease-causing variants and an early (age 3-4.5 months) disease manifestation. Analysis of Arg600His transfected COS-7 cells demonstrated less than 2% residual GAA enzyme activity compared to wild-type controls [34]. Relatively low GAA enzyme activity has also been found in patient six (P6) with c.-32-13 T > G/c.307 T > G genotype (Table 1; Figure 1A,B). In our patient, the age of onset was 32 years, although this genotype is usually linked to childhood-onset Pompe disease, with manifestation falling between 8 and 14 years of age [52]. Disease modifier genes may contribute to the phenotypic variability linked to certain genotypes [54].

In this study, we also examined the effect of alterations and their combinations on each domain of the GAA protein and α-glucosidase enzyme activity. In cases where the c.-32-13T > G variant is not present, enzyme activity is significantly lower (Figure 1B). Of the 14 missense or nonsense variants, eight (57%) are located in the catalytic CH31 (α/β)8 barrier domain, followed by three missense variants (21%) in the N-terminal β-sheet domain (Figure 2A). A similar distribution of alterations was found in a recent study in which 245 reported missense variants were mapped. In all, 64% of all pathogenic missense variants involve the catalytic CH31 (α/β)8 barrier domain, 22% fall to the N-terminal β-sheet domain and the remaining 14% fall to the other three domains [55]. Applying X-ray crystallography, the effect of different genotypes of Pompe disease to protein structure can be modelled. The findings indicate that Pompe disease is primarily a protein folding disorder, as mutations tend to disrupt the hydrophobic core of the protein [55].

The structural and conformational changes of the GAA protein as a result of different alteration is currently less extensively studied. The localization of different disease-causing variants affecting the GAA gene manifests in distinct protein domain alterations, which correlates to the level of GAA enzyme activity and CRIM status, affecting phenotypes. All of these have a main effect on disease progression through not only intracellular glycogen accumulation but also through lipofuscin formation. A better understanding of the pathogenic effects of GAA variants may highlight the importance of early initialization of ERT treatment, even if symptoms are still subtle [13,56]. Further analyses of the specific GAA variations may also provide more information on their possible development of a deleterious immune response or on the efficacy of treatment.

## Figures and Tables

**Figure 1 life-11-00507-f001:**
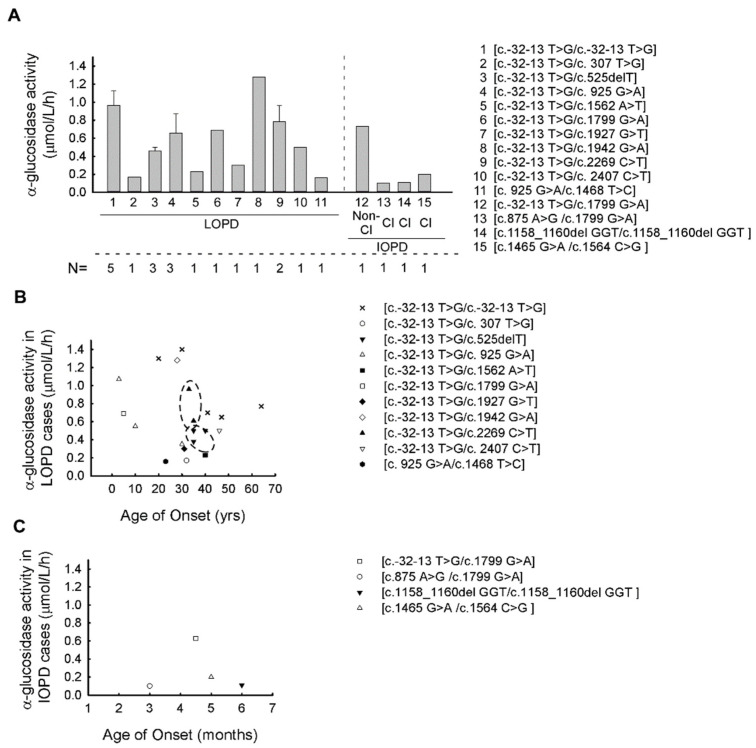
Association between GAA enzyme activity and patient’s genotypes in LOPD and IOPD: (**A**) The average α-glucosidase enzyme activity in different GAA genotypes in our cohort (the average for each genotype was calculated from the average values of the individual patient’s enzyme activity). (**B**) Correlation between the α-glucosidase enzyme activity and age of onset in different GAA genotypes in LOPD cases (genotypes showing high association with age of onset are indicated by a dashed line—the trends observed in these genotypes ([c.-32-13 T > G/c.525delT] and [c.-32-13 T > G/c.2269 C > T]) using the Pearson correlation did not show any significant differences). (**C**) Correlation between the α-glucosidase enzyme activity and age of onset in different GAA genotype in IOPD cases. (Abbreviations: LOPD—late-onset Pompe disease; IOPD—infantile-onset Pompe disease; Non-Cl—non-classical, Cl—classical, N—number of affected patients).

**Figure 2 life-11-00507-f002:**
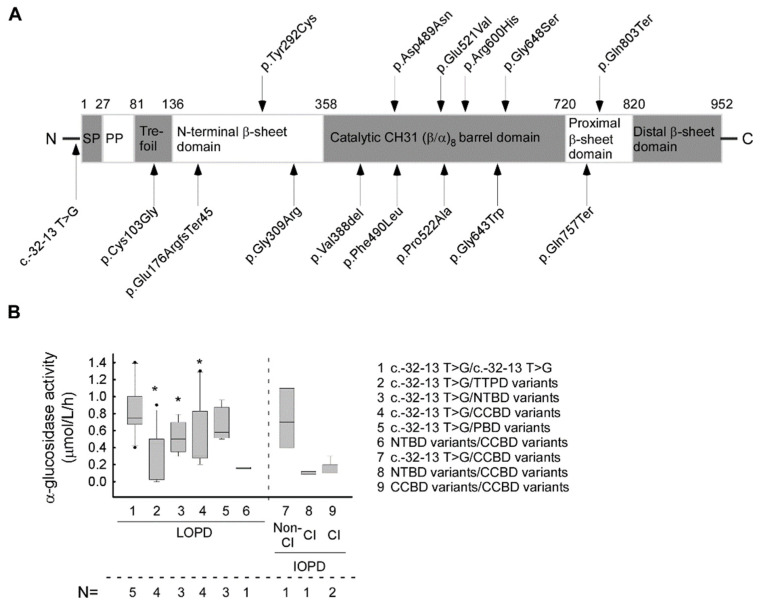
The localization of the found GAA mutations in protein structure and their effect on α-glucosidase enzyme activity: (**A**) Schematic representation of localization of the GAA mutations in protein structure. (**B**) The average α-glucosidase enzyme activity linked to different protein domain involvement in our LOPD/IOPD cohort (data analyses includes all 3 available enzyme activity values from all patients). (Abbreviations: SP—signal peptide; PP—propeptide, LOPD—late-onset Pompe disease; IOPD—infantile-onset Pompe disease; Cl—classical; non-Cl—non classical; TTP domain—trefoil type-P domain; NTB domain—N-terminal β-sheet domain; CCB domain—catalytic CH31 (α/β)8 barrel domain; PB domain—proximal β-sheet domain; N=—number of affected patients). (* *p* < 0.05). The lines represent the median values. To calculate significance, the groups LOPD 2, 3, 4 and 5 were compared to LOPD group 1 [c.-32-13 C > T/c.-32-13 C > T]. In the case of group LOPD 6, because the variant only affected a single patient, significance could not be calculated.

**Table 1 life-11-00507-t001:** Detected GAA genotypes in the study cohort.

Patient Number	Family Number	Genotype	GAA Activity (μmol/L/h) (Average ± SD)	Gender	AOO (ys)	Type of PD
P1	F1	c.-32-13 T > G/c.-32-13 T > G	1.40 ± 0.1	f	30	LOPD
P2	F1	c.-32-13 T > G/c.-32-13 T > G	1.30 ± 0.15	m	20	LOPD
P3	F2	c.-32-13 T > G/c.-32-13 T > G	0.65 ± 0.23	f	47	LOPD
P4	F2	c.-32-13 T > G/c.-32-13 T > G	0.77 ± 0.15	f	64	LOPD
P5	F3	c.-32-13 T > G/c.-32-13 T > G	0.70 ± 0.1	m	41	LOPD
P6	F4	c.-32-13 T > G/c.307 T > G	0.17 ± 0.2	f	32	LOPD
P7	F5	c.-32-13 T > G/c.525delT	0.50 ± 0.1	f	35	LOPD
P8	F5	c.-32-13 T > G/c.525delT	0.50 ± 0.2	f	40	LOPD
P9	F6	c.-32-13 T > G/c.525delT	0.38 ± 0.41	f	35	LOPD
P10	F7	c.-32-13 T > G/c.925 G > A	1.07 ± 0.23	m	3	LOPD
P11	F7	c.-32-13 T > G/c.925 G > A	0.55 ± 0.05	f	10	LOPD
P12	F8	c.-32-13 T > G/c.925 G > A	0.35 ± 0.08	f	30	LOPD
P13	F9	c.-32-13 T > G/c.1562 A > T	0.23 ± 0.08	f	40	LOPD
P14	F10	c.-32-13 T > G/c.1799 G > A	0.69 ± 0.55	m	5	LOPD
P15	F11	c.-32-13 T > G/c.1799 G > A	0.73 ± 0.15	m	0.37	IOPD
P16	F12	c.-32-13 T > G/c.1927G > T	0.30 ± 0	m	31	LOPD
P17	F13	c.-32-13 T > G/c.1942 G > A	1.28 ± 0.12	m	28	LOPD
P18	F14	c.-32-13 T > G/c.2269 C > T	0.96 ± 0.23	f	33	LOPD
P19	F14	c.-32-13 T > G/c.2269 C > T	0.61 ± 0.1	f	35	LOPD
P20	F15	c.-32-13 T > G/c.2407 C > T	0.50 ± 0.07	m	46	LOPD
P21	F16	c.875 A > G/c.1799 G > A	0.10 ± 0.05	f	0.25	IOPD
P22	F17	c.925 G > A/c.1468 T > C	0.16 ± 0.13	f	23	LOPD
P23	F18	c.1158_1160del GGT/c.1158_1160del GGT	0.11 ± 0.02	f	0.5	IOPD
P24	F19	c.1465 G > A/c.1564 C > G	0.20 ± 0.1	f	0.4	IOPD

(Abbreviations: f—female, m—male, LOPD—late-onset Pompe disease, IOPD—infant-onset Pompe disease, SD—standard deviation).

**Table 2 life-11-00507-t002:** Detected GAA disease-causing alterations in the study cohort.

No	Nt. Change	AA Change	Affec-ted Allele Number	Domain	Rsid	ACMG Score	Gnomad AF	Mutation Type	Pathogenety Scores by Vasrsome Database	Predic-ted Severity	Phenoty-pe with Null Allele	CRIM Status	Reference
1	c.-32-13 T > G	-	25	-	rs386834236	P	<0.001	splice site	1P	PM	C/A	pos	[30,31]
2	c. 307 T > G	p.Cys103Gly	1	TTPD	rs398123174	P	<0.001	missense	19P/3B	PLS	CI	pos	[32]
3	c.525delT	p.Glu176Arg fsTer45	3	TTPD	rs386834235	P	<0.001	INDEL	na	VS	CI	neg	[33]
4	c.875A > G	p.Tyr292Cys	1	NTBD	rs1057516600	P/LP	<0.001	missense	20P/1B	PM	CI	pos	[32,34]
5	c.925 G > A	p.Gly309Arg	4	NTBD	rs543300039	P	<0.001	missense	20P/0B	PLS	CI	pos	[35,36]
6	c.1158_1160del GGT	p.Val388del	2	CCBD	na	LP	na	INDEL	na	na	na	pos	ps
7	c.1465G > A	p.Asp489Asn	1	CCBD	rs398123169	P/LP	<0.001	missense	20P/1B	PLS	CI	pos	[37]
8	c.1468 T > C	p.Phe490Leu	1	CCBD	na	LP	na	missense	21P	LS	unkn	pos	[11]
9	c.1562 A > T	p.Glu521Val	1	CCBD	rs1455277014	LP	<0.001	missense	20P/1B	unkn	CI	unk	[38]
10	c.1564C > G	p.Pro522Ala	1	CCBD	rs892129065	P/LP	<0.001	missense	20P/1B	PLS	CI	pos	[39]
11	c.1799 G > A	p.Arg600His	3	CCBD	rs377544304	LP	<0.001	missense	19P/1B	PLS	CI	pos	[39]
12	c.1927G > T	p.Gly643Trp	1	CCBD	rs28937909	P	<0.001	missense	20P/1B	na	na	na	[32]
13	c.1942 G > A	p.Gly648Ser	1	CCBD	rs536906561	P	<0.001	missense	19P/1B	PLS	CI	unk	[40]
14	c.2269 C > T	p.Gln757Ter	2	PBD	rs200483245	P	<0.001	nonsense	4P/4B	VS	CI	neg	[41]
15	c. 2407 C > T	p.Gln803Ter	1	PBD	rs1344266804	LP	<0.001	nonsense	7P/1B	VS	unkn	unk	[42]

(Abbreviations: TTPD—trefoil type-P domain; NTBD—N-terminal β-sheet domain; CCBD—catalytic CH31 (α/β-)8 barrel domain; PBD—proximal β—sheet domain; P—pathogenic; LP—likely pathogenic; na—non-available; B—benign; PM—potentially mild; PLS—potentially less severe; VS—very severe; unk—unknown; C—childhood; A—adult; CI—classic infantile; pos—positive; neg—negative; ps—present study; ACMG—American College of Medical Genetics).

## Data Availability

The data presented in this study are available on request from the corresponding author.

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
