# Peer review of "Correlation of GAA Genotype and Acid-α-Glucosidase Enzyme Activity in Hungarian Patients with Pompe Disease"

_life, 2021, doi:10.3390/life11060507_

Round 1

Reviewer 1 Report

The authors describe 24 Hungarian patients of Pompe disease. They find 15 different mutations and distributed over 14 genotypes. One mutation has not been described before. GAA activity of all patients has been determined in dried blood sample extracts according to a previously described method. The authors attempt correlation of GAA activity with age of disease onset, genotype, and mutation localization within the protein. A general correlation of GAA activity and age of onset could be confirmed. More specific effects of individual mutations on GAA activity and age of onset are investigated and discussed, but solid statistical analyses are hindered by small number of patients in several genotype groups. This manuscript is generally well structured with regard to the text. The figure/table order should be improved, as well as the data presentation and annotation, in order to assess the validity of the data. The findings are certainly a relevant addition to the Pompe disease research field.

Major concerns:

The authors have ‘Determination of CRIM status’ in the method section claiming that patient 23 CRIM status has been analyzed. In Table 2 the Status is given as ‘non-available’. In line 181 the authors speak of a western blot. Please show the Western blot. It should contain CRIM-positive and CRIM-negative control.

The commercial kit ‘SALSA MLPA P453 GAA probemix’ was used to detect quantitative changed in GAA genomic sequence. This kit requires validation with DNA from healthy individuals. Have healthy individuals been included in the study? Please describe how this has been done. Also, it is not clear where data on the quantitative changes in the GAA gene have been used in the manuscript. Please clarify and show the data.

The determination of enzyme activities in blood surely depends on the extraction and storage conditions of the biological specimen. Please describe, when and at which locations the blood was drawn, how it was conserved until activity analyses, and how that is in line with previous precautions mentioned in publications.

It is unclear how many dried blood spots were measured per patient and for each CDC quality control material. Please provide assay validation data (positive and negative control, QC material). In Table 1 ‘average GAA activity’ is given, please add the standard deviation as an indicator of technical variability. Also, the unit of GAA activity should be mentioned in the table.

The authors should explicitly name the elements of the box plots in Figure 1 (what do line, box, error bars represent?). Why are some individual values shown and some not? Panels A, B, and C would greatly benefit from indicating the number of patients per bar/box, for instance by showing ALL patients as individual data points.

In figure 1C all 24 patients are grouped into onset category and genotypes. Around 10 of these 15 genotype groups are only represented by ONE patient. Please explain what the error bar represents in these ONE-patient-genotypes. Clearly it is not the biological variability between patients with the same genotype. The number of patients per group should be clearly indicated, and in cases were only one patient is represented showing error bars may not be meaningful but are rather misleading.

Line 206-214: The authors explain the effects of mutations in different protein domains on GAA activity. The protein domains are shown in Fig 2. Then they refer back to figure 1B for the correlation with the GAA activity. I think it would be better to bring the data in 1B as 2B (just beneath the protein schematic). Also, it is not entirely clear what groups were compared when the authors come to the conclusion that there were significant changes. Please indicate significant changes and compared groups (and patient numbers in each group) in Figure 1B.

The mean GAA activity of patients P6, P7, P8, P9 carrying [c.-32-13 T>G/TTPD] mutations is given as 0.367±0.293 in the text, but the average line in the box plot is clearly above 0.4. What is correct, the text or the graph?

Line 216-217: By what statistical evaluation do the authors support their assessment of a ‘very strong association’? I find it difficult to see any correlations in Figure 1D. Maybe the authors should prepare separate figures for subgroups of patients were associations are studied with statistical means.

One patient shows relatively high GAA activity but has a very early onset. The authors classify that patient as ‘atypical’. Please discuss how this contradictory finding could be explained.

Minor concerns:

Line 56: Should be “If A mutation results IN the absence…”

Line 56-58: please reword/clarify – unclear what is meant/referred to by “accounting as CRIM negative status”.

Line 58: “if there is a decreased amount of enzyme synthetized due to the mutation”. This sentence should include mutations that do not affect the amount of GAA synthesized but only its stability or function. This could be fixed by removing the word ‘synthetized’.

Line 59-60: “the recombinant enzyme replacement does not induce significant immune response and is classified as CRIM positive”. The patient should be classified as CRIM positive or negative, not the enzyme replacement.

Line 69: “leads TO impaired…”

Line 70-77: provide adequate citations for the individual intracellular processes that are thought to occur in GAA-deficiency (e.g. evidence for lysosomal rupture, problems of excess cytosolic glycogen, glycogen-induced cell death, autophagy dysregulation, protein misfolding, trafficking, and lipofuscin generation in correlation with GAA deficiency). Citation [12] does not provide a connection of lipofuscin with glycogen or Pompe disease. Maybe citation [14] could be used in addition.

Line 79: should be “ beside THE PRIMARY lysosomal pathology”

Line 86: should be “LATER-initiated” not “latter-initiated”

Line 177: “1-1 families”?

In the text there is consistently a space before the unit ‘mol/L/h’. Please confirm that this is the correct unit.

Line 198: alpha/beta is missing within the parenthesis. It seems the text does not show any greek letters.

Line 208Figure 1B. Please indicate significant changes.

Line 217: c.525deT should be c.525delT

Line 206: there is no figure 2B.

Table1, patient 18. What does C3>T mean?

Order of figures/tables could be adjusted to facilitate reading. I would prefer Table1, Table2, Figure1 (without B), Figure 2 (with data from what is in 1B now).

Line 319: beginning, not beginning

Author Response

Response to Reviewer 1 Comments

Major concerns:

Point1: The authors have ‘Determination of CRIM status’ in the method section claiming that patient 23 CRIM status has been analyzed. In Table 2 the Status is given as ‘non-available’. In line 181 the authors speak of a western blot. Please show the Western blot. It should contain CRIM-positive and CRIM-negative control.

Response1: In Table 2 CRIM status of P23 was corrected as it is positive. Determination of CRIM status of P23 was peformed by the Laboratory of Department of Chemical Pathology, Great Ormond Street Hospital for Children NHS Trust, as stated in line 136-139. We only have the report on it. The laboratory is accredited by UKAS by ISO15189:2012. The scope of accreditation can be found on the UKAS website (https://www.ukas.com/search-accredited-organisations/).

Point2: The commercial kit ‘SALSA MLPA P453 GAA probemix’ was used to detect quantitative changed in GAA genomic sequence. This kit requires validation with DNA from healthy individuals. Have healthy individuals been included in the study? Please describe how this has been done. Also, it is not clear where data on the quantitative changes in the GAA gene have been used in the manuscript. Please clarify and show the data.

Response2: For each run of the MLPA assay, a minimum of 2 healthy controls were used as reference. The coffalyser application (MRC Holland) is used to calculate possible GAA exon deletions and duplications. During the genetic analyses of GAA mutations, MLPA test was performed routinely on all patients. In our patients exon duplications and deletions in the GAA gene were not detected in any of the cases. This was mentioned in the revised manuscript result section in line 162-163.

Point3: The determination of enzyme activities in blood surely depends on the extraction and storage conditions of the biological specimen. Please describe, when and at which locations the blood was drawn, how it was conserved until activity analyses, and how that is in line with previous precautions mentioned in publications.

Response3: Preparation, storage and shipping of DBSs was carried in line with precautions previously (Elbin et al., Clinica Chimica Acta, 2011). Dried blood spots were collected by finger pricking with appropriate lancets followed by application of blood drops to the filter paper card. Samples were dried at ambient temperature and humidity. Transport to laboratory was carried out in paper envelopes within 4 days following blood sampling, whereafter prior analysis DBS samples were stored at -20 C in plastic bags with dessicant.

The sentence “Preparation, storage and shipping of DBSs were carriedout in line with previous precautions [26]” was added.

Point4: It is unclear how many dried blood spots were measured per patient and for each CDC quality control material. Please provide assay validation data (positive and negative control, QC material). In Table 1 ‘average GAA activity’ is given, please add the standard deviation as an indicator of technical variability. Also, the unit of GAA activity should be mentioned in the table.

Response4: In Table 1 the unit of GAA activity was added to the header. Response validation data are summarised below:

Calculation based on 496 analysed samples

GAA activity (mmol/L/h)

mean

14,5

median

13,8

5 percentile

2,0

95 percentile

27,4

2 percectile

0,6

98 percentile

35,1

CDC QC material (lot nr)

enzyme activity (mmol/L/h)

CDC analytical information

mean (95%CL)

mean (min-max) measured activity

SD

low (4-2009)

0,9 (0,6-1,2)

1,3 (1-1,7)

0,2

medium (4-2009)

6,8 (5,1-8,6)

9,7 (7,7-10,8)

1,3

high (4-2009)

13,3 (11,1-15,5)

18,7 (15,5-23,3)

2,9

positive control

n.a.

0,06 (0,02-0,08)

0,0

Point5: The authors should explicitly name the elements of the box plots in Figure 1 (what do line, box, error bars represent?). Why are some individual values shown and some not? Panels A, B, and C would greatly benefit from indicating the number of patients per bar/box, for instance by showing ALL patients as individual data points.

Response5: In this graph the vertical box plot style was used. In this plot the median, 10th 25th, 75th and 90th percentiles have been graphed as vertical boxes with error bars. The lines represent the average values, the error bar represents the standard error. During the data analysis all available enzyme activity values were used from every patient. For genotypes that occurred in only one patient, the standard deviation was calculated using each dried blood spot samples.  The individual data points for all patients were shown in Table 1.

Point6: In Figure 1C all 24 patients are grouped into onset category and genotypes. Around 10 of these 15 genotype groups are only represented by ONE patient. Please explain what the error bar represents in these ONE-patient-genotypes. Clearly it is not the biological variability between patients with the same genotype. The number of patients per group should be clearly indicated, and in cases were only one patient is represented showing error bars may not be meaningful but are rather misleading.

Response6: In this figure (old version: Figure 1C, new version: Figure 1A) the error bars represent the standard deviation of GAA enzyme activities measured (at least two in every patient). During the data analysis all available enzyme activity values were included from every patients. For genotypes that occurred in only one patient, the standard deviation was calculated using each dried blood spot samples.  This was mentioned in the reversed manuscript in the Figure 1 legend section in line 242-249

Point7: Line 206-214: The authors explain the effects of mutations in different protein domains on GAA activity. The protein domains are shown in Fig 2. Then they refer back to figure 1B for the correlation with the GAA activity. I think it would be better to bring the data in 1B as 2B (just beneath the protein schematic). Also, it is not entirely clear what groups were compared when the authors come to the conclusion that there were significant changes. Please indicate significant changes and compared groups (and patient numbers in each group) in Figure 1B.

Response7: Thank you for your suggestion, we moved Figure 1B to Figure 2B. When calculating the significancy of the changes, certain mutation combinations were compared to c.-32-13 T>G homozygous cases. This is indicated in the text below: “By comparing the a-glucosidase enzyme activity of c.-32-13 T>G homozygous and compound heterozygous cases, the mean GAA activity in homozygous form is significantly higher than that of the compound heterozygous cases (p=0.003)” (line 210-215). The patient numbers in each group are shown in Figure 2B.

Point8: The mean GAA activity of patients P6, P7, P8, P9 carrying [c.-32-13 T>G/TTPD] mutations is given as 0.367±0.293 in the text, but the average line in the box plot is clearly above 0.4. What is correct, the text or the graph?

Response8: The correct version was included in the new version of the manuscript. There was a slip in the sigmaplot file for unknown reason, but it also returned the specified value as an average. The error was corrected in Figure 2B.

Point9: Line 216-217: By what statistical evaluation do the authors support their assessment of a ‘very strong association’? I find it difficult to see any correlations in Figure 1D. Maybe the authors should prepare separate figures for subgroups of patients were associations are studied with statistical means.

Response9: The genotypes indicating high association are indicated by a dashed line. This is written in Figure 1 legend in line  242-249.

Point10: One patient shows relatively high GAA activity but has a very early onset. The authors classify that patient as ‘atypical’. Please discuss how this contradictory finding could be explained.

Response10: Thank you for your suggestion. The term ‘atypical’ was replaced by non-classical infantile type in both the text and the figure, as well. In some cases broad phenotype variability can be seen with the same genotype which raises the possible impact of modifying gene effect. (Kroos et al, Neurology. 2007 Jan 9;68(2):110-5)  This explanation was implemented into the new version (in line 359-360).

Minor concerns:

Point11: Line 56: Should be “If A mutation results IN the absence…”

Response11:  This section was rephrased as “Patients are classified into CRIM (cross reactive immunologic material) negative and CRIM positive groups, depending on whether the enzyme is completely absent (CRIM-) or present (CRIM+) in lower amounts.”

Point 12-13-14:

Line 56-58: please reword/clarify – unclear what is meant/referred to by “accounting as CRIM negative status”.

Line 58: “if there is a decreased amount of enzyme synthetized due to the mutation”. This sentence should include mutations that do not affect the amount of GAA synthesized but only its stability or function. This could be fixed by removing the word ‘synthetized’.

Line 59-60: “the recombinant enzyme replacement does not induce significant immune response and is classified as CRIM positive”. The patient should be classified as CRIM positive or negative, not the enzyme replacement.

Response12-13-14: The sentence about CRIM status in line 59-61 was modified as “Patients are classified into CRIM (cross reactive immunologic material) negative and CRIM positive groups, depending on whether the enzyme is completely absent (CRIM-) or present in lower amounts (CRIM+).”

Point15: Line 69: “leads TO impaired…”

Response15: The text was corrected to “Decreased activity leads TO impaired…”

Point16: Line 70-77: provide adequate citations for the individual intracellular processes that are thought to occur in GAA-deficiency (e.g. evidence for lysosomal rupture, problems of excess cytosolic glycogen, glycogen-induced cell death, autophagy dysregulation, protein misfolding, trafficking, and lipofuscin generation in correlation with GAA deficiency). Citation [12] does not provide a connection of lipofuscin with glycogen or Pompe disease. Maybe citation [14] could be used in addition.

Response16:

Additional references were included to support the pathological processes in Pompe disease:

  • Shimada Y, Kobayashi H, Kawagoe S, Aoki K, Kaneshiro E, Shimizu H, Eto Y, Ida H, Ohashi T. Endoplasmic reticulum stress induces autophagy through activation of p38 MAPK in fibroblasts from Pompe disease patients carrying c.546G>T mutation. Mol Genet Metab. 2011 Dec;104(4):566-73. doi: 10.1016/j.ymgme.2011.09.005. Epub 2011 Sep 14. PMID: 21982629. (Protein misfolding)
  • Meikle PJ, Yan M, Ravenscroft EM, Isaac EL, Hopwood JJ, Brooks DA. Altered trafficking and turnover of LAMP-1 in Pompe disease-affected cells. Mol Genet Metab. 1999 Mar;66(3):179-88. doi: 10.1006/mgme.1998.2800. PMID: 10066386. (Impaired intracellular trafficking)
  • Fukuda T, Roberts A, Ahearn M, Zaal K, Ralston E, Plotz PH, Raben N. Autophagy and lysosomes in Pompe disease. Autophagy. 2006 Oct-Dec;2(4):318-20. doi: 10.4161/auto.2984. Epub 2006 Oct 5. PMID: 16874053.
  • Raben N, Roberts A, Plotz PH. Role of autophagy in the pathogenesis of Pompe disease. Acta Myol. 2007 Jul;26(1):45-8. PMID: 17915569; PMCID: PMC2949326. (Autophagy)

Point17: Line 79: should be “ beside THE PRIMARY lysosomal pathology”

Response17: The text was changed to “..,beside THE PRIMARY lysosomal…”.

Point18: Line 86: should be “LATER-initiated” not “latter-initiated”

Response18: The phrase was corrected to “..and LATER-INITIATED…”.

Point19: Line 177: “1-1 families”?

Response19: The listed genotypes occured in one family each.  These are the following: The c.307 T>G (p.Cys103Gly) mutation was found in Patient 6 (family 4); c.1465G>A (p.Asp489Asn) mutation was found in Patient 24 (family 19); the c.1562 A>T (p.Glu521Val) mutation was found in Patient 13 (family 9); the c.1564 C>G (p.Pro522Ala) mutation was found in Patient 24 (family 19); c.1927 G>T (p.Gly643Trp) mutation was found in Patient 16 (family 12); c.1942 G>A (p.Gly648Ser) mutation was found in Patient 17 (family 13); the c.2269 C>T (p.Gln757Ter) mutation was found in one sister pair in Patient 18, 19 (family 14); and the c.2407 C>T (p.Gln803Ter) was found in Patient 20 (family 15). (as shown in Table 2 as well).

The sentence was modified to clarify this issue.  “Another 8 mutations (c.307 T>G (p.Cys103Gly); c.1465G>A (p.Asp489Asn); c.1562 A>T (p.Glu521Val); c.1564 C>G (p.Pro522Ala); c.1927 G>T (p.Gly643Trp); c.1942 G>A (p.Gly648Ser); c.2269 C>T (p.Gln757Ter); and c.2407 C>T (p.Gln803Ter)) occurred in one family each (Table 1).” (line 182-186)

Point20: In the text there is consistently a space before the unit ‘mol/L/h’. Please confirm that this is the correct unit.

Response20: The unit of the a-glucosidase activity is mmol/L/h. Unfortunately, due to an unclear technical issue certain symbols were lost during the uploading process. It was corrected in all places.

Point21: Line 198: alpha/beta is missing within the parenthesis. It seems the text does not show any greek letters.

Response21: Thank you for your comment. Unfortunately, due to an unclear technical issue certain symbols were lost during the uploading process. It was corrected in all places.

Point22: Line 208Figure 1B. Please indicate significant changes.

Response22: Thank you for your comment. We indicated the significant changes in this figure (old version Figure 1B, new version figure 2B).

Point23: Line 217: c.525deT should be c.525delT

Response23: In line 225 the correction was made to “c.525delT”.

Point24: Line 206: there is no figure 2B.

Response24: Thank you for your comment. It was originally figure 1B but in the new version became figure 2B.

Point25: Table1, patient 18. What does C3>T mean?

Response25: Thank you for your comment. This was corrected to C>T.

Point26: Order of figures/tables could be adjusted to facilitate reading. I would prefer Table1, Table2, Figure1 (without B), Figure 2 (with data from what is in 1B now).

Response26: Thank you for your suggestion, we modified it for better reading,

Point27: Line 319: beginning, not beginning

Response27: In line 318 the word “beginning” was corrected.

Reviewer 2 Report

Authors had better present further information of patient symptoms and discussion on peudodeficienciy.

Author Response

Response to Reviewer 2 Comments

Point1: Authors had better present further information of patient symptoms and discussion on peudodeficienciy.

Response1: Thank you for your suggestion!  We included a paragraph in the new Introduction (line 56-59).  “ Lower a-glucodidase activity can be the result of the presence of an allele resulting in pseudodeficiency. The pseudodeficient allele never cause Pompe disease, this is the reason why genetic testing have to be also performed during the diagnostic processs of Pompe disease”.

In this study we focused on which protein-domains are affected by different pathogenic mutations and how these combinations of pathogenic mutations possibly affect the level of enzyme activity. In this respect, we considered the classification of patients into disease subgroups to be sufficient and did not discuss the detailed phenotype.

Reviewer 3 Report

Gal and colleagues describe in “Correlation of GAA genotype and acid-α-glucosidase enzy-matic activity and its impact on domain structure among Hungarian Pompe disease patients” a first enzymologic and genetic analysis of 24 Pompe patients in Hungary. Such data is important as it expands and fortifies the available (genetic) data in Pompe disease bases, which is essential to increase the knowledge, in particular on rare diseases like Pompe disease.

Overall the data is well described and presented, although the text will benefit from language editing. Also symbols appear to have been lost in translation to pdf. Editing errors like extra spaces need to be corrected.

Below I have listed a number of questions and comments that should be clarified by the authors:

General: The authors should acquaint them with the guidelines on reporting genomic variants according human genome variation society (HGVS) nomenclature, to which more journal adhere. The term mutation is no longer allowed and should be specified: e.g. disease-associated variant, etc. According to these rules the c.1158_1160del (line 178) should be annotated as c.1163_1165del. But indeed is a novel variant.

-The authors most likely have access to clinical data of the patients but seem to reserve these for another manuscript, while this manuscript might benefit from that with respect to genotype-phenotype correlation. The link of genotype with enzyme activity is not so novel and already available from other sources for most of the variants.

title:

the title suggests that a throrough biochemical study into protein domain function is done, which is not. Therefore the title is misleading and should be adapted.

Abstract:

The term LOPD (late onset Pompe disease) is misleading and in my view incorrect, as it refers in some studies to patients from birth –that have not the classic-infantile phenotype- to advantaged age. Also in this study the LOPD group includes patients from 3 and 5 yo and that is confusing. I feel, and there is a body of correspondence on this matter, it is more clear for both people in- and out-side the field to discriminate classic-infantile, childhood/juvenile onset and adult onset patients. A rough genotype-phenotype correlation exist between these groups, in particular with residual GAA activity. As the clinical parameters associated with these three groups are different (but overlapping), it is more clear if one describes a juvenile vs an adult onset patient. As the childhood-onset patients are still actively growing, it is not scientifically correct to group them with the adult-onset patients. This grouping also clearly discriminates patients with classic-infantile disease (IOPD) and children with a non-classic presentation (that have a completely different prognosis). The “atypical”patient (P15) with residual activity in the range of adult-onset patients, with the c.-32-13T>G variant, will not have the severe classic-infantile phenotype. The term “atypical” should be replaced by non-classic infantile Pompe disease. The term IOPD is therefore, just as LOPD  confusing and incorrect to group this patient with the other patients that do have the severe classic infantile Pompe disease.

-Line 33: “alpha” symbol is missing->throughout text there are symbols missing. I stopped indicating them after this. Please check the pdf that is generated before submitting.

Introduction

-CRIM status: It is unclear why a relatively large part of the introduction is dedicated to CRIM status as it is not a major factor of this work. In the end only 3 of the reported patients could have been CRIM-negative, 2 of these are suspected CRIM-positive according to the public databases. CRIM status is important for prognosis and should be determined for the group of classic infantile patients, but as clinical data and assessment is not part of this study, it should not be emphasized that much in the introduction. It is misleading readers into expecting a study addressing CRIM status. It should be noted that only a part of the IOPD (with complete absence of GAA protein expression) is expected to be CRIM-negative, some estimate this to be 20% of infantile patients. This is a rough estimate and may not be cited just to indicate this not the majority of IOPD patients. 

-Line 64 I would add “… that characterize IOPD patients” after “cardiac symptoms…” to emphasize to outsiders that the cardiac involvement is not a general symptom of Pompe disease, but restricted to infantile patients.

-Line 66-73 description of cellular pathology the Thurberg study should be acknowledged (Thurberg, B. L. et al. (2006) ‘Characterization of pre- and post-treatment pathology after enzyme replacement therapy for Pompe disease.’, Laboratory investigation; a journal of technical methods and pathology, 86(12), pp. 1208–1220. doi: 10.1038/labinvest.3700484.) which exactly describes this for the first time and included a graphical representation of this.

-Line 118: The Pompe variant database seem to be a rich source for this paper, but is not cited properly: Niño, MY, in 't Groen, SL, Bergsma, AJ, et al. Extension of the Pompe mutation database by linking disease‐associated variants to clinical severity. Human Mutation. 2019; 40: 1954–1967. https://doi.org/10.1002/humu.23854

-Line 169-170: Interesting observation, maybe worth checking this patient for reported genetic modifiers.

-Line 171 atypical IOPD is also known as non-classic infantile Pompe disease

-Line 181: a Western blot is mentioned. Why is this not shown?

-Line 259-261: the observation that enzyme activity in classic infantile Pompe patients –called IOPD in this work- is lower than in juvenile/adult onset (referred to as LOPD) is not a surprising or novel finding and is known for decades (see for instance:  Reuser et al, 1995 doi: 10.1002/mus.880181414.)

-Line 271 there is no such thing as classical adult phenotype spectrum. Replace by adult onset Pompe disease spectrum or something similar.

-Line 293 and further starting with: “This difference may be due…” needs to be removed as is scientifically incorrect. Due to the (c.-32-13T>G) variant exon 2 is not missing, but the resulting RNA is misspliced.

-Line 312 should be rephrased as the Pompe database (line 313) is also a public database and everything in there is also in the literature, so line 312 is inherently incorrect.. Change to something like: The c.1468 T>C (p.Phe490Leu) is not reported in some of the public databases like ClinVar or HGMD.

-Line 324: maybe refer to the relevance of modifier genes explaining large difference in disease onset.

-Line 340-341 is grammatically incorrect, words missing: “…is currently not studied extensively less studied”??

 -Line 343-345 is difficult to understand. Rephrase and split into two.

-Line 346: the decision making process: I guess the authors refer to start and stop criteria for ERT. There are European guidelines on that which seem to be facilitating the decision process rather well. Do the authors suggest that the mentioned factors are not taken into account into those? Please specify this.

Author Response

Response to Reviewer 3 Comments

Title

Point1: the title suggests that a throrough biochemical study into protein domain function is done, which is not. Therefore the title is misleading and should be adapted.

Response1: The title was adapted accordingly to “Correlation of GAA genotype and acid-α-glucosidase enzyme activity in Hungarian patients with  Pompe disease”.

Abstact

Point2: The term LOPD (late onset Pompe disease) is misleading and in my view incorrect, as it refers in some studies to patients from birth –that have not the classic-infantile phenotype- to advantaged age. Also in this study the LOPD group includes patients from 3 and 5 yo and that is confusing. I feel, and there is a body of correspondence on this matter, it is more clear for both people in- and out-side the field to discriminate classic-infantile, childhood/juvenile onset and adult onset patients. A rough genotype-phenotype correlation exist between these groups, in particular with residual GAA activity. As the clinical parameters associated with these three groups are different (but overlapping), it is more clear if one describes a juvenile vs an adult onset patient. As the childhood-onset patients are still actively growing, it is not scientifically correct to group them with the adult-onset patients. This grouping also clearly discriminates patients with classic-infantile disease (IOPD) and children with a non-classic presentation (that have a completely different prognosis). The “atypical”patient (P15) with residual activity in the range of adult-onset patients, with the c.-32-13T>G variant, will not have the severe classic-infantile phenotype. The term “atypical” should be replaced by non-classic infantile Pompe disease. The term IOPD is therefore, just as LOPD confusing and incorrect to group this patient with the other patients that do have the severe classic infantile Pompe disease.

Response2: In our point of view, in line with the terminology widely used, Pompe disease is classified as infantile, early onset (IOPD) and late onset (LOPD) form. This classification and approach takes into account the distinctly different phenotypes, residual enzyme activity, prognosis, etc. these two groups exhibit. Although, we indeed think LOPD patients should be referred to as childhood/juvenile onset LOPD and adult onset LOPD based on the age of onset. Also IOPD patient can further subdivided to classical and non-classical form. The text has been revised and corrected accordingly.  The term ‘atypical’ was replaced by ‘non-classic infantile’. (lines 48-56).

Point3: -Line 33: “alpha” symbol is missing->throughout text there are symbols missing. I stopped indicating them after this. Please check the pdf that is generated before submitting.

Response3: The ‘alpha’ symbol was lost during the conversion to pdf. format. The symbol was reset.

Introduction

Point4:-CRIM status: It is unclear why a relatively large part of the introduction is dedicated to CRIM status as it is not a major factor of this work. In the end only 3 of the reported patients could have been CRIM-negative, 2 of these are suspected CRIM-positive according to the public databases. CRIM status is important for prognosis and should be determined for the group of classic infantile patients, but as clinical data and assessment is not part of this study, it should not be emphasized that much in the introduction. It is misleading readers into expecting a study addressing CRIM status. It should be noted that only a part of the IOPD (with complete absence of GAA protein expression) is expected to be CRIM-negative, some estimate this to be 20% of infantile patients. This is a rough estimate and may not be cited just to indicate this not the majority of IOPD patients.

Response4: We agree with the comment pointing out the discussion on the CRIM status was over-represented, therefore we simplified this part. The sentence about CRIM status in line 59-61 was modified as “Patients are classified into CRIM (cross reactive immunologic material) negative and CRIM positive groups, depending on whether the enzyme is completely absent (CRIM-) or present in lower amounts (CRIM+).”

Point5: Line 64 I would add “… that characterize IOPD patients” after “cardiac symptoms…” to emphasize to outsiders that the cardiac involvement is not a general symptom of Pompe disease, but restricted to infantile patients.

Response5: Your suggestion was included as “..(one of the main contributor to mortality in the classical IOPD)”.

Point6: Gal and colleagues describe in “Correlation of GAA genotype and acid-α-glucosidase enzy-matic activity and its impact on domain structure among Hungarian Pompe disease patients” a first enzymologic and genetic analysis of 24 Pompe patients in Hungary. Such data is important as it expands and fortifies the available (genetic) data in Pompe disease bases, which is essential to increase the knowledge, in particular on rare diseases like Pompe disease.

Overall the data is well described and presented, although the text will benefit from language editing. Also symbols appear to have been lost in translation to pdf. Editing errors like extra spaces need to be corrected.

Response6: Unfortunately, due to an unclear technical issue certain symbols were lost during the uploading process. These and extra spaces were corrected again in the pdf format.

Point7: Below I have listed a number of questions and comments that should be clarified by the authors:

General: The authors should acquaint them with the guidelines on reporting genomic variants according human genome variation society (HGVS) nomenclature, to which more journal adhere. The term mutation is no longer allowed and should be specified: e.g. disease-associated variant, etc. According to these rules the c.1158_1160del (line 178) should be annotated as c.1163_1165del. But indeed is a novel variant.

Response7: During the data analyses based on the ClinVar and Ensemble genome browser, as a reference we used the GRCh38.p13, ENST00000302262.8, NM_000152.5 sequence. The c.1163_1165del position is calculated from reference GRCh37.p13 not the GRCh38.p13 which was used as a reference in this study. For better understanding, the GRCh38.p13 reference was entered in the methods section. (in line 110)

Point8: -The authors most likely have access to clinical data of the patients but seem to reserve these for another manuscript, while this manuscript might benefit from that with respect to genotype-phenotype correlation. The link of genotype with enzyme activity is not so novel and already available from other sources for most of the variants.

Response8: In this study we focused on which protein-domains are affected by different pathogenic mutations and how these combinations of pathogenic mutations possibly affect the level of enzyme activity. In this respect, we considered the classification of patients into disease subgroups to be sufficient and did not discuss the detailed phenotype.

Point9:-Line 66-73 description of cellular pathology the Thurberg study should be acknowledged (Thurberg, B. L. et al. (2006) ‘Characterization of pre- and post-treatment pathology after enzyme replacement therapy for Pompe disease.’, Laboratory investigation; a journal of technical methods and pathology, 86(12), pp. 1208–1220. doi: 10.1038/labinvest.3700484.) which exactly describes this for the first time and included a graphical representation of this.

Response9: We included the work Thurberg et al., which is indeed well-known and was missing.

Point10-Line 118: The Pompe variant database seem to be a rich source for this paper, but is not cited properly: Niño, MY, in 't Groen, SL, Bergsma, AJ, et al. Extension of the Pompe mutation database by linking disease‐associated variants to clinical severity. Human Mutation. 2019; 40: 1954–1967. https://doi.org/10.1002/humu.23854

Response10: We added the proper citation to the reference list.

Point11-Line 169-170: Interesting observation, maybe worth checking this patient for reported genetic modifiers.

Response11: Thank you for the comment on this observation, we also think it is worth further analysis.

Point12--Line 171 atypical IOPD is also known as non-classic infantile Pompe disease

Response12: The term ‘atypical IOPD’ was changed to ‘non-classical IOPD’.

Point13--Line 181: a Western blot is mentioned. Why is this not shown?

Response13: Determination of CRIM status of P23 was made with diagnostic purpose by the Laboratory of Department of Chemical Pathology, Great Ormond Street Hospital for Children NHS Trust, as stated in line 141-143. We do not have in hand the WB but only the report. The laboratory is accredited by UKAS against ISO15189:2012. The scope of accreditation can be found on the UKAS website (https://www.ukas.com/search-accredited-organisations/)

Point14-Line 259-261: the observation that enzyme activity in classic infantile Pompe patients –called IOPD in this work- is lower than in juvenile/adult onset (referred to as LOPD) is not a surprising or novel finding and is known for decades (see for instance:  Reuser et al, 1995 doi: 10.1002/mus.880181414.)

Response14: Thank you for your suggestion. We reconsidered figures and removed the original Figure 1A because we agree with your comment, it fails to contain new information.  

Point15-Line 271 there is no such thing as classical adult phenotype spectrum. Replace by adult onset Pompe disease spectrum or something similar.

Response15: In line 272 the term ‘classical adult phenotype spectrum’ was changed to „..the late-onset, adult phenotype.”

Point16-Line 293 and further starting with: “This difference may be due…” needs to be removed as is scientifically incorrect. Due to the (c.-32-13T>G) variant exon 2 is not missing, but the resulting RNA is misspliced.

Response16: The word ‘missing’ was replaced by ‘misspliced’ (line 327).

Point17-Line 312 should be rephrased as the Pompe database (line 313) is also a public database and everything in there is also in the literature, so line 312 is inherently incorrect.. Change to something like: The c.1468 T>C (p.Phe490Leu) is not reported in some of the public databases like ClinVar or HGMD.

Response17: The proposal was carried over the text as “..reported in the literature and some of the public databases like ClinVar or HGMD).” We wrote in the text the following: “ The c.1468 T>C (p.Phe490Leu) is currently not reported in some of the public databases like ClinVar or HGMD.” (In line 345-346)

Point18-Line 324: maybe refer to the relevance of modifier genes explaining large difference in disease onset.

Response18: An additional sentence was added “Disease modifier genes may contribute the phenotypic variability linked to certain genotypes.” In line 359-360

Point19-Line 340-341 is grammatically incorrect, words missing: “…is currently not studied extensively less studied”??

Response19: The phrase was corrected to, “..as a result of of different mutations is currently less extensively studied .” In line 375-376.

 Point20-Line 343-345 is difficult to understand. Rephrase and split into two.

Response20: Thank you for your suggestion. This sentence was split into two part. „All of these have a main effect on disease progression throughout not only intracellular glycogen accumulation but also lipofuscin formation. This can be reversed only marginally by ERT”. In line 378-380.

Point21-Line 346: the decision making process: I guess the authors refer to start and stop criteria for ERT. There are European guidelines on that which seem to be facilitating the decision process rather well. Do the authors suggest that the mentioned factors are not taken into account into those? Please specify this.

Response21: It was meant by this sentence to emphasize and stress out the importance of early treatment initialization if symptoms occur. The sentence was rephrased to clarify as “ A comprehensive approach evaluating these factors stress out the importance of early initialization of ERT treatment if symptoms occur even they subtle. Additional information also may be provided on the possible development of an immune response to, and the efficacy of treatment.” Lines 381-384.

Round 2

Reviewer 1 Report

I appreciate that the authors have achieved significant improvement of their manuscript. Many of my previous concerns were sufficiently addressed. In these cases I did not add a comment in blue font below the authors’ responses. Occasionally, there are small changes required, that I think will further improve the quality of the manuscript. I am still concerned with the statistical analyses in Fig. 1A and 2B. The revised version is better than before, but there are still issues with respect to technical vs. biological replicates, average and SD calculation, and data representation that need to be addressed before publication. Please find my detailed responses in the attached docx file.

Author Response

Response to Reviewer 1 Comments

I appreciate that the authors have achieved significant improvement of their manuscript. Many of my previous concerns were sufficiently addressed. In these cases I did not add a comment in blue font below the authors’ responses. Occasionally, there are small changes required, that I think will further improve the quality of the manuscript. I am still concerned with the statisical analyses in Fig. 1A and 2B. The revised version is better than before, but there are issues with respect to technical vs. biological replicates, average and SD calculation, and data representation that need to be fixed before publication.

Major concerns:

Point1: The authors have ‘Determination of CRIM status’ in the method section claiming that patient 23 CRIM status has been analyzed. In Table 2 the Status is given as ‘non-available’. In line 181 the authors speak of a western blot. Please show the Western blot. It should contain CRIM-positive and CRIM-negative control.

Response1: In Table 2 CRIM status of P23 was corrected as it is positive. Determination of CRIM status of P23 was peformed by the Laboratory of Department of Chemical Pathology, Great Ormond Street Hospital for Children NHS Trust, as stated in line 136-139. We only have the report on it. The laboratory is accredited by UKAS by ISO15189:2012. The scope of accreditation can be found on the UKAS website (https://www.ukas.com/search-accredited-organisations/).

The accreditation should be added to the method section.

Response1B: We added the following sentence in the methods section. „The laboratory is accredited by UKAS by ISO15189:2012.” Line 150

The certificate on the accreditation is included in the reference list [29].

Point2: The commercial kit ‘SALSA MLPA P453 GAA probemix’ was used to detect quantitative changed in GAA genomic sequence. This kit requires validation with DNA from healthy individuals. Have healthy individuals been included in the study? Please describe how this has been done. Also, it is not clear where data on the quantitative changes in the GAA gene have been used in the manuscript. Please clarify and show the data.

Response2: For each run of the MLPA assay, a minimum of 2 healthy controls were used as reference. The coffalyser application (MRC Holland) is used to calculate possible GAA exon deletions and duplications. During the genetic analyses of GAA mutations, MLPA test was performed routinely on all patients. In our patients exon duplications and deletions in the GAA gene were not detected in any of the cases. This was mentioned in the revised manuscript result section in line 162-163.

The use healthy controls should be mentioned in the method section.

Response2B: The following sentence was added in the method section „For the normalization 3 different healthy control samples were used per each run” Line 119-120

Point3: The determination of enzyme activities in blood surely depends on the extraction and storage conditions of the biological specimen. Please describe, when and at which locations the blood was drawn, how it was conserved until activity analyses, and how that is in line with previous precautions mentioned in publications.

Response3: Preparation, storage and shipping of DBSs was carried in line with precautions previously (Elbin et al., Clinica Chimica Acta, 2011). Dried blood spots were collected by finger pricking with appropriate lancets followed by application of blood drops to the filter paper card. Samples were dried at ambient temperature and humidity. Transport to laboratory was carried out in paper envelopes within 4 days following blood sampling, whereafter prior analysis DBS samples were stored at -20 C in plastic bags with dessicant.

The sentence “Preparation, storage and shipping of DBSs were carriedout in line with previous precautions [26]” was added.

Line 133: should be ’carried OUT’ (in the manuscript OUT is missing)

Response3B: The manuscript was corrected accordingly as, ’carried out.’ (line 141)

Point4: It is unclear how many dried blood spots were measured per patient and for each CDC quality control material. Please provide assay validation data (positive and negative control, QC material). In Table 1 ‘average GAA activity’ is given, please add the standard deviation as an indicator of technical variability. Also, the unit of GAA activity should be mentioned in the table.

Response4: In Table 1 the unit of GAA activity was added to the header. Response validation data are summarised below:

Calculation based on 496 analysed samples

GAA activity (mmol/L/h)

mean

14,5

median

13,8

5 percentile

2,0

95 percentile

27,4

2 percectile

0,6

98 percentile

35,1

I have difficulties to understand this response. The paper describes results form 24 patients. The above table calculates statistics over 496 samples. How are these 496 samples related to the 24 patients in the paper?

If the 496 samples were measured in healthy controls to determine the ’normal’ range of GAA activity, then this should be stated in the method section. The normal range is then not ’on the basis of the 95th percentile’, but rather ’>2 umol/L/h (greater than 5th percentile)’.

If, however, for each patient multiple assays were performed. In my opinion it is not meaningful to calculate a mean/median/etc across all 496 assays. Rather for each patient, an average value should be calculated, which derives only form those few assays performed on the one patient. The average activity determined for each patient can then be analyzed with respect to other patients and CDC control material.

CDC QC material (lot nr)

enzyme activity (mmol/L/h)

CDC analytical information

mean (95%CL)

mean (min-max) measured activity

SD

low (4-2009)

0,9 (0,6-1,2)

1,3 (1-1,7)

0,2

medium (4-2009)

6,8 (5,1-8,6)

9,7 (7,7-10,8)

1,3

high (4-2009)

13,3 (11,1-15,5)

18,7 (15,5-23,3)

2,9

positive control

n.a.

0,06 (0,02-0,08)

0,0

How come, the positive control of the QC has the lowest activity? Is this a negative control?

Response4B: 

The 496 samples, which we referred to in our previous answer, were measured previous to this investigation. The blood spot samples from all cases were suspected of having different lysosomal storage disorders and healty controls were tested with this multiplex assay (determination of GLA, GBA, GAA, MSPS1, ABG enzyme activities simultaneously).

Since most of these cases have normal GAA activity compared to the positive controls, the 5th percentile was calculated based on the total previously measured 496 samples.

Assay validation data consists of CDC QC material measurements (2nd table in reponse4), through which we proved that the used assay is appropriate for GAA activity measurements for unknown samples. In case of enzyme activity measurements, the positive quality control means low enzyme activity (defect in enzyme activity), which is the case in the 2nd table. Negative control is not provided by CDC since all healthy/wild type individuals will present normal GAA activity. The statistics presented in the 1st table as regard to measurements of 496 samples, positive range is defined as lower than 5th percentile (2mmol/L/h). This data was provided only as response to Point4, and was not meant to be included in the manuscript. Since the laboratory evidence consists of mean enzyme activity (3 samples, each measured in triplicates) of a given patient with no SD, and the purpose of the current study is not focused on development of a GAA activity assay, in Table 1 only this value is given.

To avoid misunderstanding, in the manuscript Table 1. header „average GAA activity” is changed to „GAA activity (mmol/L/h)”

To avoid misunderstanding, in the manuscript Table 1. header „average GAA activity” is changed to „GAA activity (mmol/L/h)”

Point5: The authors should explicitly name the elements of the box plots in Figure 1 (what do line, box, error bars represent?). Why are some individual values shown and some not? Panels A, B, and C would greatly benefit from indicating the number of patients per bar/box, for instance by showing ALL patients as individual data points.

Response5: In this graph the vertical box plot style was used. In this plot the median, 10th 25th, 75th and 90th percentiles have been graphed as vertical boxes with error bars. The lines represent the average values, the error bar represents the standard error. During the data analysis all available enzyme activity values were used from every patient. For genotypes that occurred in only one patient, the standard deviation was calculated using each dried blood spot samples.  The individual data points for all patients were shown in Table 1.

In the Box plot (now Fig. 2B), was median or average used? Why are there two individual data points shown for group 1 LOPD and one indivdidual data point for group 4 LOPD? Please clarify. The description of graph elements should be integrated in the figure legend. 

In one graph statistical parameters should either evaluate technical variation (i.e. the variation of repeated determinations of the same patient), or to evaluate biological variation (i.e. the variation of activities across several patients). In graph 1A and 2B statistical parameters (SD, percentiles etc.) are sometimes coming form technical repeats (same patient) and sometimes from biological repeats (several patients). This is uncommon, misleading, and should be avoided

I think it is beneficial to evaluate the technical variation for each patient AND to also get an idea of the biological variation across several patients. I would therefore ask the following: 1) for each of the 24 patients, calculate the average activity across all repeated assay for this one patient. 2) This average should be listed in Table 1. 3) Add to table 1 the SD across assays within the same patient as an expression of the technical variation. 4) Do not show percentiles or error bars in groups with N=1 in Fig. 1A and 2B. Just show the average activity for this one patient. 5) Keep the description of N below the graphs (this is helpful). 6) adjust the figure legend with respect to the changes.

In the case of group 6 LOPD, statistical comparison with Shapiro-Wilk is not meaningful because there is only ONE sample. It is very important to perform the statistical analyses only after averages for each patient have been calculated. E.g. for comparing groups 1 and 2 LOPD, there are five average values from five patients in one group and four average values from four patients in the other group. The average of the five patients in group 1 is then statistically compared with the average of four patients in group 2.

Response5B: 

We double checked again the diagram structure of the boxplot graph. We might misunderstand your comment. The lines represent the median values. We mentioned it in the figure legend.  The individual data points represent the outliers. 

For the calculation of the average values, we used only the biological repeats. For each patient, we had the results of GAA enzyme activity from at least 3 repeated blood spots, which can be considered biological parallels in our view. Each result is derived from 3 technical replicates, which can thus be considered a biological replicate.

.

Figure 1A was redesigned accordingly to your recommendation. The average for each genotype were calculated from the average values of the individual patient’s enzyme activity. Thus, for genotypes that represented by only one patient, the figure does not include SD. This was mentioned in the Figure1 legend “ Line 265-265

SD values were added to Table1.

On Figure2B all available enzyme activity from patients were used in the data analysis process. These did not include technical repeats, only the biological parallels were used.

Thank you for your comment. The significance mark for LOPD 6 was taken from Figure 2B. 

Point6: In Figure 1C all 24 patients are grouped into onset category and genotypes. Around 10 of these 15 genotype groups are only represented by ONE patient. Please explain what the error bar represents in these ONE-patient-genotypes. Clearly it is not the biological variability between patients with the same genotype. The number of patients per group should be clearly indicated, and in cases were only one patient is represented showing error bars may not be meaningful but are rather misleading.

Response6: In this figure (old version: Figure 1C, new version: Figure 1A) the error bars represent the standard deviation of GAA enzyme activities measured (at least two in every patient). During the data analysis all available enzyme activity values were included from every patient. For genotypes that occurred in only one patient, the standard deviation was calculated using each dried blood spot samples.  This was mentioned in the reversed manuscript in the Figure 1 legend section in line 242-249

Same as above: In graph 1A statistical parameters (SD etc.) are sometimes coming form technical repeats (same patient) and sometimes from biological repeats (several patients). This is uncommon, misleading, and should be avoided. During data analysis it is inappropriate to calculate an average and SD across all technical repetitions AND several patients as this leads to false representation of the data. Please see the example calculation below. From 3 fictitious patients values were determined with 2 to 5 technical repetitions per patient. In F4 and G4 average and SD was calculated across all values in the cells ranging from C4 to E8. Both average and SD are different from the values in F9 and G9. Here average and SD was calculated from the three averages in C9, D9, and E9. (C9, D9, and E9 were calculated for each patient from the technical repeats for the same patient.) The way, F9 and G9 are calculated is the appropriate way of analysis.

I do think it is beneficial to evaluate the technical variation for each patient AND to also get an idea of the biological variation across several patients. I would therefore ask the following: 1) for each of the 24 patients, calculate the average activity across all repeated assay for this one patient. 2) This average should be listed in Table 1 (as it is). 3) Add to table 1 the SD across assays within the same patient as an expression of the technical variation. 4) Do not show error bars in groups with N=1 in Fig. 1A. Just show the average activity for this one patient. 5) Keep the description of N below the graphs (this is helpful).

Response6B: 

In all cases 3 biological  replicates were evaluated from 3 different blood spots. All blood spots were measured 3 times (technical replicates). This description was added into the methods section. For our calculation we used 3 reported values of the biological replicates. Each value is derived from 3 separeted measurement of the given sample which corresponds to technical replicates. It means SDs are coming from the measured values of the biological replicates.

On data analysis we calculated with the available activity values. For each measurement point (values from a blood collection card), one value was used. These values are reported by the Laboratory of the First Department of Pediatrics, Semmelweis University. These values are derived from the 3 measured technical replicates. 

Point7: Line 206-214: The authors explain the effects of mutations in different protein domains on GAA activity. The protein domains are shown in Fig 2. Then they refer back to figure 1B for the correlation with the GAA activity. I think it would be better to bring the data in 1B as 2B (just beneath the protein schematic). Also, it is not entirely clear what groups were compared when the authors come to the conclusion that there were significant changes. Please indicate significant changes and compared groups (and patient numbers in each group) in Figure 1B.

Response7: Thank you for your suggestion, we moved Figure 1B to Figure 2B. When calculating the significancy of the changes, certain mutation combinations were compared to c.-32-13 T>G homozygous cases. This is indicated in the text below: “By comparing the a-glucosidase enzyme activity of c.-32-13 T>G homozygous and compound heterozygous cases, the mean GAA activity in homozygous form is significantly higher than that of the compound heterozygous cases (p=0.003)” (line 210-215). The patient numbers in each group are shown in Figure 2B.

Line 214: change to „higher than THAT of the compound heterozygous cases”

Response7: The referred part was changed in the text to „higher than THAT” Line 225

Point8: The mean GAA activity of patients P6, P7, P8, P9 carrying [c.-32-13 T>G/TTPD] mutations is given as 0.367±0.293 in the text, but the average line in the box plot is clearly above 0.4. What is correct, the text or the graph?

Response8: The correct version was included in the new version of the manuscript. There was a slip in the sigmaplot file for unknown reason, but it also returned the specified value as an average. The error was corrected in Figure 2B.

Point9: Line 216-217: By what statistical evaluation do the authors support their assessment of a ‘very strong association’? I find it difficult to see any correlations in Figure 1D. Maybe the authors should prepare separate figures for subgroups of patients were associations are studied with statistical means.

Response9: The genotypes indicating high association are indicated by a dashed line. This is written in Figure 1 legend in line  242-249.

The authors still did not explain how they determined ’association’. The two genotype groups they marked in Fig 1B contain 3 and 2 patients, respectively. 4 of these 5 patients have the same onset age, with the fifth being also very close. By contrast, the GAA activity level varies between 0.3 and 1. That rather indicates that in these groups GAA activity is INDEPENDENT of age of onset, because one parameter changes a lot while the other remains stable. If a correlation is stated, please calculate for instance Pearson coefficients and show them in the paper. The sentence should be changed to somthing like „In the LOPD group, very strong association of genotype with age of onset has been found in [c.-32-13 T>G/c.525delT], [c.-32-13 T>G/c.2269 C>T] genotypes (Figure 1B).” Note, that the geneotypes only associate with regard to age of onset, not with GAA activity. Therefore, GAA and age of onset does NOT associate. This should be explained more clearly in the results part.

Response9B: In the LOPD group, very similar age of onset and GAA activity values were found with the [c.-32-13 T>G/c.525delT] and [c.-32-13 T>G/c.2269 C>T] genotypes (Figure 1B). In case of the [c.-32-13 T>G/c.525delT] genotype in all 3 affected subjects (P7, P8, P9), closely clus-tered values were indicated for both age of onsets and GAA activities (age of onsets are: 35ys, 40 yrs. and 35 yrs.; GAA activities are: 0,5, 0,5 and 0,38 μmol/L/h respectively). In P18 and P19 patients the genotype [c.-32-13 T>G/c.2269 C>T] was associated with a very similar age of onset (33 and 35 yrs.), but less close GAA activity values. The trends, ob-served in the two genotypes ([c.-32-13 T>G/c.525delT] and [c.-32-13 T>G/c.2269 C>T]) did not show any significant difference using the Pearson correlation (Figure 1B). It was corrected in the result part and the figure legend.  (Line 237-246; Line 266-268)

Point10: One patient shows relatively high GAA activity but has a very early onset. The authors classify that patient as ‘atypical’. Please discuss how this contradictory finding could be explained.

Response10: Thank you for your suggestion. The term ‘atypical’ was replaced by non-classical infantile type in both the text and the figure, as well. In some cases broad phenotype variability can be seen with the same genotype which raises the possible impact of modifying gene effect. (Kroos et al, Neurology. 2007 Jan 9;68(2):110-5)  This explanation was implemented into the new version (in line 359-360).

Minor concerns:

Point11: Line 56: Should be “If A mutation results IN the absence…”

Response11:  This section was rephrased as “Patients are classified into CRIM (cross reactive immunologic material) negative and CRIM positive groups, depending on whether the enzyme is completely absent (CRIM-) or present (CRIM+) in lower amounts.”

Point 12-13-14:

Line 56-58: please reword/clarify – unclear what is meant/referred to by “accounting as CRIM negative status”.

Line 58: “if there is a decreased amount of enzyme synthetized due to the mutation”. This sentence should include mutations that do not affect the amount of GAA synthesized but only its stability or function. This could be fixed by removing the word ‘synthetized’.

Line 59-60: “the recombinant enzyme replacement does not induce significant immune response and is classified as CRIM positive”. The patient should be classified as CRIM positive or negative, not the enzyme replacement.

Response12-13-14: The sentence about CRIM status in line 59-61 was modified as “Patients are classified into CRIM (cross reactive immunologic material) negative and CRIM positive groups, depending on whether the enzyme is completely absent (CRIM-) or present in lower amounts (CRIM+).”

Point15: Line 69: “leads TO impaired…”

Response15: The text was corrected to “Decreased activity leads TO impaired…”

Point16: Line 70-77: provide adequate citations for the individual intracellular processes that are thought to occur in GAA-deficiency (e.g. evidence for lysosomal rupture, problems of excess cytosolic glycogen, glycogen-induced cell death, autophagy dysregulation, protein misfolding, trafficking, and lipofuscin generation in correlation with GAA deficiency). Citation [12] does not provide a connection of lipofuscin with glycogen or Pompe disease. Maybe citation [14] could be used in addition.

Response16:

Additional references were included to support the pathological processes in Pompe disease:

  •       Shimada Y, Kobayashi H, Kawagoe S, Aoki K, Kaneshiro E, Shimizu H, Eto Y, Ida H, Ohashi T. Endoplasmic reticulum stress induces autophagy through activation of p38 MAPK in fibroblasts from Pompe disease patients carrying c.546G>T mutation. Mol Genet Metab. 2011 Dec;104(4):566-73. doi: 10.1016/j.ymgme.2011.09.005. Epub 2011 Sep 14. PMID: 21982629. (Protein misfolding)
  •       Meikle PJ, Yan M, Ravenscroft EM, Isaac EL, Hopwood JJ, Brooks DA. Altered trafficking and turnover of LAMP-1 in Pompe disease-affected cells. Mol Genet Metab. 1999 Mar;66(3):179-88. doi: 10.1006/mgme.1998.2800. PMID: 10066386. (Impaired intracellular trafficking)
  •       Fukuda T, Roberts A, Ahearn M, Zaal K, Ralston E, Plotz PH, Raben N. Autophagy and lysosomes in Pompe disease. Autophagy. 2006 Oct-Dec;2(4):318-20. doi: 10.4161/auto.2984. Epub 2006 Oct 5. PMID: 16874053.
  •       Raben N, Roberts A, Plotz PH. Role of autophagy in the pathogenesis of Pompe disease. Acta Myol. 2007 Jul;26(1):45-8. PMID: 17915569; PMCID: PMC2949326. (Autophagy)

Point17: Line 79: should be “ beside THE PRIMARY lysosomal pathology”

Response17: The text was changed to “..,beside THE PRIMARY lysosomal…”.

Point18: Line 86: should be “LATER-initiated” not “latter-initiated”

Response18: The phrase was corrected to “..and LATER-INITIATED…”.

Point19: Line 177: “1-1 families”?

Response19: The listed genotypes occured in one family each.  These are the following: The c.307 T>G (p.Cys103Gly) mutation was found in Patient 6 (family 4); c.1465G>A (p.Asp489Asn) mutation was found in Patient 24 (family 19); the c.1562 A>T (p.Glu521Val) mutation was found in Patient 13 (family 9); the c.1564 C>G (p.Pro522Ala) mutation was found in Patient 24 (family 19); c.1927 G>T (p.Gly643Trp) mutation was found in Patient 16 (family 12); c.1942 G>A (p.Gly648Ser) mutation was found in Patient 17 (family 13); the c.2269 C>T (p.Gln757Ter) mutation was found in one sister pair in Patient 18, 19 (family 14); and the c.2407 C>T (p.Gln803Ter) was found in Patient 20 (family 15). (as shown in Table 2 as well).

The sentence was modified to clarify this issue.  “Another 8 mutations (c.307 T>G (p.Cys103Gly); c.1465G>A (p.Asp489Asn); c.1562 A>T (p.Glu521Val); c.1564 C>G (p.Pro522Ala); c.1927 G>T (p.Gly643Trp); c.1942 G>A (p.Gly648Ser); c.2269 C>T (p.Gln757Ter); and c.2407 C>T (p.Gln803Ter)) occurred in one family each (Table 1).” (line 182-186)

Point20: In the text there is consistently a space before the unit ‘mol/L/h’. Please confirm that this is the correct unit.

Response20: The unit of the a-glucosidase activity is mmol/L/h. Unfortunately, due to an unclear technical issue certain symbols were lost during the uploading process. It was corrected in all places.

Point21: Line 198: alpha/beta is missing within the parenthesis. It seems the text does not show any greek letters.

Response21: Thank you for your comment. Unfortunately, due to an unclear technical issue certain symbols were lost during the uploading process. It was corrected in all places.

Point22: Line 208Figure 1B. Please indicate significant changes.

Response22: Thank you for your comment. We indicated the significant changes in this figure (old version Figure 1B, new version figure 2B).

Figure 2B: Please indicate in the figure legend that statistical evaluation was performed by comparing each group with group 1 LOPD (I assume that was the case). In the case of group 6 LOPD, statistical comparison with Shapiro-Wilk is not meaningful because there is only ONE sample. It is very important to perform the statistical analyses only after averages for each patient have been calculated. E.g. for comparing groups 1 and 2 LOPD, there are five average values from five patients in one group and four average values from four patients in the other group. The average of the five patients in group 1 is then statistically compared with the average of four patients in group 2. Do not mix biological and technical replicates (please see my comments above).

Response22B: We recalculated the significance levels and group 6LOPD on Figure 2B was excluded from the calculation. 

We added this sentence to the Figure 2 legend: “ To calculate significance, the groups LOPD 2, 3, 4, 5 were compared to LOPD group 1 [c.-32-13 C> T / c.-32-13 C> T].   In LOPD 6 case because of the variant only affected one patient, significance cannot be calculated”. 

We agree with you, the analyses we used only for the biological replicates. At data analysis we calculated with all of the enzyme activity available. For each measurement point (values from a blood collection card), one value was counted. These values were sent from the metabolic laboratory and were included those in the patient reports. These values are derived from the average of 3 measurements (technical replicates) but these have only been our biological parallel.

Point23: Line 217: c.525deT should be c.525delT

Response23: In line 225 the correction was made to “c.525delT”.

Point24: Line 206: there is no figure 2B.

Response24: Thank you for your comment. It was originally figure 1B but in the new version became figure 2B.

Point25: Table1, patient 18. What does C3>T mean?

Response25: Thank you for your comment. This was corrected to C>T.

Point26: Order of figures/tables could be adjusted to facilitate reading. I would prefer Table1, Table2, Figure1 (without B), Figure 2 (with data from what is in 1B now).

Response26: Thank you for your suggestion, we modified it for better reading,

Point27: Line 319: beginning, not beginning

Response27: In line 318 the word “beginning” was corrected.

Reviewer 2 Report

none

Author Response

We didn't get another question

Reviewer 3 Report

Overall the authors have clearly addressed the reviewers comments. In some cases, which are included below, the authors response was not clear or evoked additional comments.

*General: The authors should acquaint them with the guidelines on reporting genomic variants according human genome variation society (HGVS) nomenclature, to which more journal adhere. The term mutation is no longer allowed and should be specified: e.g. disease-associated variant, etc.

The authors have responded only to the second part of this comment (about the specific c.1158_1160del variant).

In short: the word mutation is according to the current nomenclature rules no longer allowed and has to be changed throughout the text. Please consult the most recent nomenclature rules. For instance consult: Higgins, J.; Dalgleish, R.; Dunnen, J.T. den; Barsh, G.; Freeman, P.J.; Cooper, D.N.; Cullinan, S.; Davies, K.E.; Dorkins, H.; Gong, L.; Imoto, I.; Klein, T.E.; Korf, B.; Misra, A.; Paalman, M.H.; Ratzel, S.; Reichardt, J.K.V.; Rehm, H.L.; Tokunaga, K.; Weck, K.E. & Cutting, G.R. (2021), Verifying nomenclature of DNA variants in submitted manuscripts: Guidance for journals, Human Mutation 42(1): 3-7.

 Or look at the HGVS website:

https://www.hgvs.org/mutnomen/

*Related to this Line 36-37 was changed into “The localization of mutations is correlated with GAA enzyme activity through protein domain involvement”., but in my view makes no sense.

I guess the authors want to express something like this: The localizations of the identified sequence variations in regions encoding for critical protein domains of GAA correlate with severe effects on enzyme activity.

*Similarly: suggestions for the last 2 sentences of the abstract:

A better understanding of the impact of the pathogenic gene variations can emphasize the need for early initiation of enzyme replacement therapy (ERT) if subtle symptoms occur. Further information on the effect of GAA gene variation  may also be provided on the efficacy of treatment and the extent of immune response to ERT would be of importance for disease management and designing effective treatment plans.

*Original comment (Point9):-Line 66-73 description of cellular pathology the Thurberg study should be acknowledged (Thurberg, B. L. et al. (2006) ‘Characterization of pre- and post-treatment pathology after enzyme replacement therapy for Pompe disease.’, Laboratory investigation; a journal of technical methods and pathology, 86(12), pp. 1208–1220. doi: 10.1038/labinvest.3700484.) which exactly describes this for the first time and included a graphical representation of this.

Response of the authors: We included the work Thurberg et al., which is indeed well-known and was missing.

New reviewer’s comment: the Thurberg paper still does not appear in the reference list and is still missing!??!

*Comment to Authors Response18. A word “to” was missing:  An additional sentence was added “Disease modifier genes may contribute to the phenotypic variability linked to certain genotypes.” In line 359-360

*Original reviewers comment to: Point20-Line 343-345 is difficult to understand. Rephrase and split into two.

Authors Response20: Thank you for your suggestion. This sentence was split into two part. „All of these have a main effect on disease progression throughout not only intracellular glycogen accumulation but also lipofuscin formation. This can be reversed only marginally by ERT”. In line 378-380.

Reviewers new comment: I think the sentence should read: „All of these have a main effect on disease progression throughout not only intracellular glycogen accumulation but also through lipofuscin formation.

*Original reviewers comment: Point21-Line 346: the decision making process: I guess the authors refer to start and stop criteria for ERT. There are European guidelines on that which seem to be facilitating the decision process rather well. Do the authors suggest that the mentioned factors are not taken into account into those? Please specify this.

Authors Response21: It was meant by this sentence to emphasize and stress out the importance of early treatment initialization if symptoms occur. The sentence was rephrased to clarify as “ A comprehensive approach evaluating these factors stress out the importance of early initialization of ERT treatment if symptoms occur even they subtle. Additional information also may be provided on the possible development of an immune response to, and the efficacy of treatment.” Lines 381-384.

*New reviewers comment: I understand what the authors intend to say, but the sentences are a bit difficult to understand. My suggestion:

A better understanding of the pathogenic effects of GAA variants may further stress the importance of early initialization of ERT treatment even if symptoms are still subtle [13,48]. Further analyses of the specific GAA variations may also provide more information on their possible development of a deleterious immune response or on the efficacy of treatment.”

Is this what the authors wanted to say?

Author Response

Response to Reviewer 3 Comments

Overall the authors have clearly addressed the reviewers comments. In some cases, which are included below, the authors response was not clear or evoked additional comments.

Point1: *General: The authors should acquaint them with the guidelines on reporting genomic variants according human genome variation society (HGVS) nomenclature, to which more journal adhere. The term mutation is no longer allowed and should be specified: e.g. disease-associated variant, etc.

The authors have responded only to the second part of this comment (about the specific c.1158_1160del variant).

In short: the word mutation is according to the current nomenclature rules no longer allowed and has to be changed throughout the text. Please consult the most recent nomenclature rules. For instance consult: Higgins, J.; Dalgleish, R.; Dunnen, J.T. den; Barsh, G.; Freeman, P.J.; Cooper, D.N.; Cullinan, S.; Davies, K.E.; Dorkins, H.; Gong, L.; Imoto, I.; Klein, T.E.; Korf, B.; Misra, A.; Paalman, M.H.; Ratzel, S.; Reichardt, J.K.V.; Rehm, H.L.; Tokunaga, K.; Weck, K.E. & Cutting, G.R. (2021), Verifying nomenclature of DNA variants in submitted manuscripts: Guidance for journals, Human Mutation 42(1): 3-7.

 Or look at the HGVS website:

https://www.hgvs.org/mutnomen/

Response1: The term of mutation has been replaced in the text and figures with synonyms which are accepted by HGMD. 

Point2: *Related to this Line 36-37 was changed into “The localization of mutations is correlated with GAA enzyme activity through protein domain involvement”., but in my view makes no sense.

I guess the authors want to express something like this: The localizations of the identified sequence variations in regions encoding for critical protein domains of GAA correlate with severe effects on enzyme activity.

Response2: It was replaced to “The localization of the identified sequence variations in regions encoding for crucial protein domains of GAA correlate with severe effects on enzyme activity. “ Line 37-38.

Point3:*Similarly: suggestions for the last 2 sentences of the abstract:

A better understanding of the impact of the pathogenic gene variations can emphasize the need for early initiation of enzyme replacement therapy (ERT) if subtle symptoms occur. Further information on the effect of GAA gene variation  may also be provided on the efficacy of treatment and the extent of immune response to ERT would be of importance for disease management and designing effective treatment plans.

Response3: We added it to the abstract . “Further information on the effect of GAA gene variation on the efficacy of treatment and the extent of immune response to ERT would be of importance for optimal disease management and designing effective treatment plans.” Line 40-42.

Point4:*Original comment (Point9):-Line 66-73 description of cellular pathology the Thurberg study should be acknowledged (Thurberg, B. L. et al. (2006) ‘Characterization of pre- and post-treatment pathology after enzyme replacement therapy for Pompe disease.’, Laboratory investigation; a journal of technical methods and pathology, 86(12), pp. 1208–1220. doi: 10.1038/labinvest.3700484.) which exactly describes this for the first time and included a graphical representation of this.

Response of the authors: We included the work Thurberg et al., which is indeed well-known and was missing.

New reviewer’s comment: the Thurberg paper still does not appear in the reference list and is still missing!??!

Response4: This reference was added to the reference list (reference 14).

Point5: :*Comment to Authors Response18. A word “to” was missing:  An additional sentence was added “Disease modifier genes may contribute to the phenotypic variability linked to certain genotypes.” In line 359-360

Response5: This sentence was added to the text. 

Point6:*Original reviewers comment to: Point20-Line 343-345 is difficult to understand. Rephrase and split into two.

Authors Response20: Thank you for your suggestion. This sentence was split into two part. „All of these have a main effect on disease progression throughout not only intracellular glycogen accumulation but also lipofuscin formation. This can be reversed only marginally by ERT”. In line 378-380.

Reviewers new comment: I think the sentence should read: „All of these have a main effect on disease progression throughout not only intracellular glycogen accumulation but also through lipofuscin formation.

Response6: This sentence was corrected based on your suggestion. (line 395-396)

Point7:*Original reviewers comment: Point21-Line 346: the decision making process: I guess the authors refer to start and stop criteria for ERT. There are European guidelines on that which seem to be facilitating the decision process rather well. Do the authors suggest that the mentioned factors are not taken into account into those? Please specify this.

Authors Response21: It was meant by this sentence to emphasize and stress out the importance of early treatment initialization if symptoms occur. The sentence was rephrased to clarify as “ A comprehensive approach evaluating these factors stress out the importance of early initialization of ERT treatment if symptoms occur even they subtle. Additional information also may be provided on the possible development of an immune response to, and the efficacy of treatment.” Lines 381-384.

*New reviewers comment: I understand what the authors intend to say, but the sentences are a bit difficult to understand. My suggestion:

A better understanding of the pathogenic effects of GAA variants may further stress the importance of early initialization of ERT treatment even if symptoms are still subtle [13,48]. Further analyses of the specific GAA variations may also provide more information on their possible development of a deleterious immune response or on the efficacy of treatment.”

Is this what the authors wanted to say?

Response7:  Yes, thank you for the correction and suggestion. This was incorporated into the text “A better understanding of the pathogenic effects of GAA variants may stress the importance of early initialization of ERT treatment even if symptoms are still subtle [13, 49]. Further analyses of the specific GAA variations may also provide more information on their possible development of a deleterious immune response or on the efficacy of treatment.”  (line 396-400)
